# Organic matter mineralization in modern and ancient ferruginous sediments

André Friese [1,15], Kohen Bauer[2,3,15], Clemens Glombitza [4,5], Luis Ordoñez[6], Daniel Ariztegui [6], Verena B. Heuer [7], Aurèle Vuillemin[1,8], Cynthia Henny[9], Sulung Nomosatryo[1,9], Rachel Simister [2,3], Dirk Wagner[1,10], Satria Bijaksana[11], Hendrik Vogel [12], Martin Melles [13], James M. Russell [14], Sean A. Crowe [2,3✉] & Jens Kallmeyer [1✉]

Deposition of ferruginous sediment was widespread during the Archaean and Proterozoic Eons, playing an important role in global biogeochemical cycling. Knowledge of organic matter mineralization in such sediment, however, remains mostly conceptual, as modern ferruginous analogs are largely unstudied. Here we show that in sediment of ferruginous Lake Towuti, Indonesia, methanogenesis dominates organic matter mineralization despite highly abundant reactive ferric iron phases like goethite that persist throughout the sediment. Ferric iron can thus be buried over geologic timescales even in the presence of labile organic carbon. Coexistence of ferric iron with millimolar concentrations of methane further demonstrates lack of iron-dependent methane oxidation. With negligible methane oxidation, methane diffuses from the sediment into overlying waters where it can be oxidized with oxygen or escape to the atmosphere. In low-oxygen ferruginous Archaean and Proterozoic oceans, therefore, sedimentary methane production was likely favored with strong potential to influence Earth's early climate.

[1] GFZ German Research Centre for Geosciences, Potsdam, Germany. [2] Department of Microbiology and Immunology, University of British Columbia, Vancouver, Canada. [3] Department of Earth, Ocean, and Atmospheric Sciences, University of British Columbia, Vancouver, Canada. [4] ETH Zürich, Institute of Biogeochemistry and Pollutant Dynamics, Zürich, Switzerland. [5] Center for Geomicrobiology, Aarhus University, Aarhus, Denmark. [6] Department of Earth Sciences, University of Geneva, Geneva, Switzerland. [7] MARUM – Center for Marine Environmental Sciences, University of Bremen, Bremen, Germany. [8] Department of Earth & Environmental Sciences, Paleontology & Geobiology, Ludwig-Maximilians-Universität München, Munich, Germany. [9] Research Center for Limnology, Indonesian Institute of Sciences (LIPI), Cibinong, Bogor, West Java, Indonesia. [10] Institute of Geosciences, University of Potsdam, Potsdam, Germany. [11] Faculty of Mining and Petroleum Engineering, Institut Teknologi Bandung, Bandung, Jawa Barat, Indonesia. [12] Institute of Geological Sciences & Oeschger Centre for Climate Change Research, University of Bern, Bern, Switzerland. [13] Institute of Geology and Mineralogy, University of Cologne, Cologne, Germany. [14] Department of Earth, Environmental, and Planetary Sciences, Brown University, Providence, RI, USA. [15] These authors contributed equally: André Friese, Kohen Bauer. ✉email: sean.crowe@ubc.ca; jens.kallmeyer@gfz-potsdam.de

Atmospheric chemistry, and its evolution over geological time, is intrinsically linked to the burial and mineralization of organic matter[1]. Burial of organic matter can be a net source of oxidants (like oxygen) as well as a net sink of $CO_2$, and its mineralization can result in production of greenhouse gasses like methane[2]. In modern marine sediments, overlain by oxygenated bottom waters with abundant sulfate, much of the organic matter mineralization proceeds via a combination of aerobic respiration and sulfate reduction[3] with methanogenesis as the terminal step of carbon mineralization[4]. More than 90% of the produced methane is consumed through anaerobic oxidation of methane (AOM) with sulfate as terminal electron acceptor precluding appreciable fluxes to the overlying water and atmosphere[5]. Oxygen exposure in these sediments also directly controls sedimentary organic carbon preservation and bottom water anoxia increases carbon burial efficiency[6]. The Precambrian ocean–atmosphere system was much different than today's—the atmosphere was only weakly oxygenated, seawater was sulfate poor, and the oceans were generally characterized by ferruginous (anoxic, Fe-rich) conditions[7,8]. Precipitation of Fe from these oceans resulted in the widespread deposition of ferruginous siliciclastic sediments—which are by definition Fe-rich and contain highly reactive Fe that is not pyritized—and, in cases, nearly pure chemical sediments like many banded iron formations (BIFs)[8]. The fate of organic matter, and biogeochemical cycling of climatically important trace gases, in Precambrian sediments is thus intrinsically linked to coupled C and Fe cycling.

In the complete, or near, absence of oxygen, nitrate and sulfate, organic matter mineralization in ferruginous sediments would be expected to proceed anaerobically via the energetically most favorable terminal electron acceptors available—in this case ferric iron, followed by $CO_2$ through methanogenesis[4]. Prior work in freshwater and wetland sediments indeed shows that iron reducing bacteria outcompete methanogens for electron donors[9], even when Fe(III) is supplied in the form of more crystalline (oxyhydr) oxides like goethite (FeOOH), provided surface area is sufficiently high to allow microbial access to Fe(III) surface sites[10]. Laboratory studies with synthetic and natural silica-rich Fe (oxyhydr) oxides and enrichment cultures further imply that Fe(III) can be effectively reduced when sufficient carbon is supplied[11,12]. The role of Fe-reduction in organic matter mineralization, however, remains largely untested in Fe(III)-rich modern ferruginous environments analogous to those of the Precambrian oceans. Studies in the permanently stratified water column of Lake Matano, Indonesia[13] suggest that, despite abundant iron, methanogenesis is responsible for up to 90% of the total anaerobic organic matter mineralization. Process rates, however, were not measured in the Lake Matano study, whereas the enrichment culture experiments[11,12] were conducted in the laboratory under conditions that deviate considerably from likely environmental conditions. The role of methanogenesis, in both modern and ancient ferruginous sediments therefore remains largely untested through direct measurements in the natural environment.

We recovered modern ferruginous sediments from Lake Towuti, Indonesia, and used a suite of biogeochemical analyses to directly determine rates and pathways of organic matter mineralization. Lake Towuti is situated on Sulawesi Island, has a maximum water depth of 203 m (Fig. 1) and is weakly thermally stratified with a well-mixed, oxygenated surface layer that extends to 70 m depth, and waters below 130 m that are persistently anoxic[14]. Intensive weathering of ophiolitic bedrock from the catchment supplies the lake with a strong influx of iron (oxyhydr) oxides and runoff that contains little sulfate, leading to sulfate poor (<20 μM) lake water and anoxic ferruginous conditions with Fe(II) concentrations up to 40 μM below 130 m[14]. Similar conditions were also reported in nearby Lake Matano, which is considered broadly analogous to Precambrian ferruginous oceans[15]. The Fe (oxyhydr)oxide flux to Lake Towuti is comprised mainly of poorly- to nanocrystalline goethite with lesser amounts of hematite and magnetite, all of which may have been reworked and recrystallized to some extent during transport or through redox cycling in the uppermost sediments before burial[16].

As part of the Towuti Drilling Project (TDP) of the International Scientific Drilling Program (ICDP), we recovered sediment from a water depth of 156 m (Fig. 1)[17], well below the oxycline at the time of sampling[18]. This drill core was supplemented with short (<0.4 m) gravity cores that better preserve the sediment–water interface (SWI). Radiocarbon dating revealed a nearly constant sedimentation rate of 19 cm ka$^{-1}$, yielding an estimated age of ~60 ka at 12 m depth[19].

## Results

**Sediment and pore water geochemistry.** Lake Towuti's sediment is relatively rich in organic carbon (TOC 0.4–4 wt%) with elemental compositions implying that it is reactive (molar C:N ratio 11–25), readily fermentable (Fig. 2), and consequently reactive towards microbial respiration. Given a sedimentation rate of 19 cm ka$^{-1}$ [19] and a bulk density of 1.3 g cm$^{-3}$, these organic

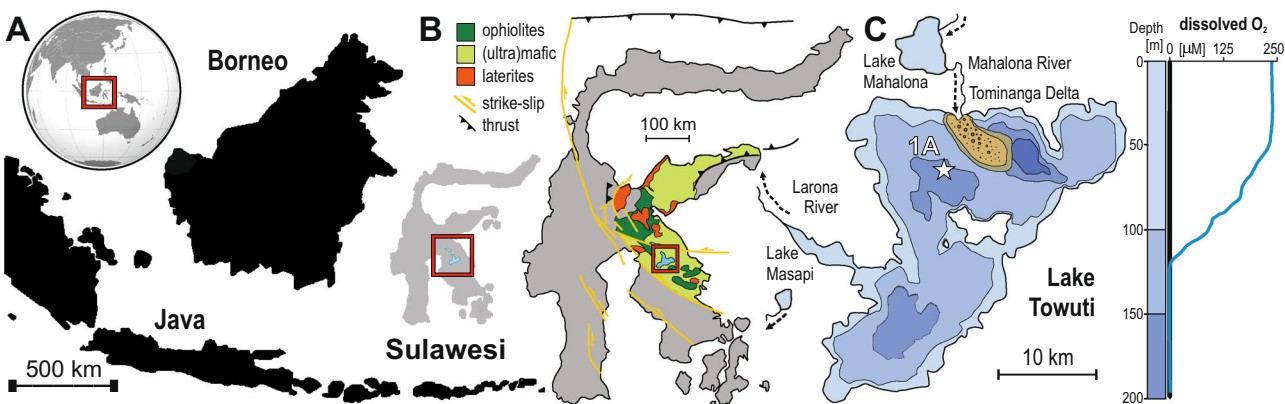

**Fig. 1 Lake Towuti location and geological setting. A** Map of Indonesia and the location of Sulawesi Island. **B** Map of Sulawesi Island illustrating the geological setting of the Malili Lake system, a chain of five tectonic lakes, with Lake Towuti being the largest. **C** Bathymetric map of Lake Towuti, the white star marks the location of the drilling site 1A, which is located at a water depth of 153 m, well below the depth of the oxycline (130 m). Figure modified after Vuillemin et al. (2016)[14].

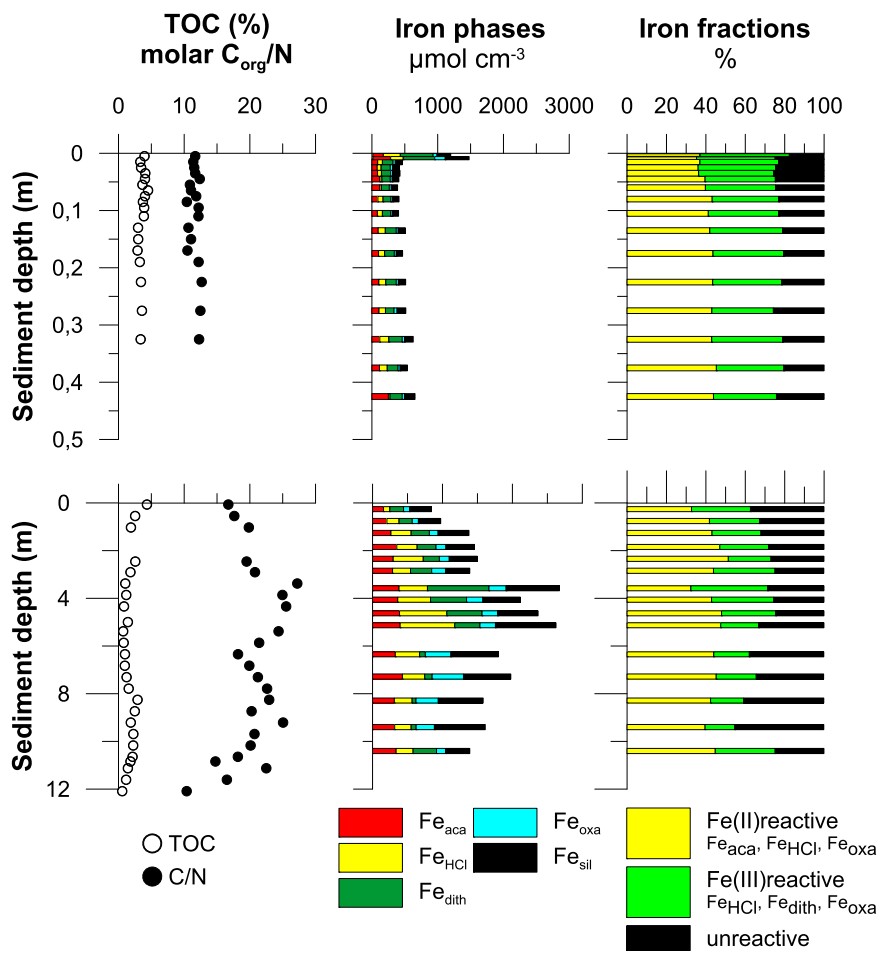

**Fig. 2 Sediment characteristics of the upper 12 m of the drill core recovered at Site 1 in Lake Towuti.** Total organic carbon content in bulk sediment and molar $C_{org}/N$ ratio (left rows), sequentially extracted iron phases (center rows) and iron fractions (right rows) of Lake Towuti sediment of the short (<0.4 m) gravity core (upper panel) and the 12 m TDP drill core (lower panel). See main text and supplementary material for detailed description of the iron phases. Total organic carbon content as well as molar $C_{org}/N$ ratios of the short gravity core were taken from Vuillemin et al. (2016)[14]. Note, that Fe(III) within the $Fe_{HCl}$ fraction was below our limit of detection so that the reactive Fe(III) pool is composed entirely of the $Fe_{dith}$ and $Fe_{oxa}$ fraction. Error bars were omitted for clarity, for TOC and C/N ratio they are smaller than the symbols. See Table S5 for iron extraction data, including error margins.

carbon concentrations translate to organic carbon accumulation rates between 220 and 1800 mmol m$^{-2}$ yr$^{-1}$, which thus places an upper bound on rates of total sedimentary carbon respiration. Fermentation is a key step in organic matter mineralization and its main products are volatile fatty acids (VFA) and molecular hydrogen, which are known electron donors for iron reduction[20]. We detected formate, acetate, lactate, propionate, and butyrate (Fig. 3), all showing the highest concentrations in the upper 2–6 m. Ammonium, which is also a product of organic matter mineralization[4], reaches peak pore water concentrations around 6 m (Fig. 3). DIC concentrations range from 2 to 4 mM, indicating that the system is well buffered with respect to pH, which thus has a limited range of 6.8–7.2 (Fig. 4). Taken together, these results indicate that microbial degradation of organic matter takes place throughout the sediment, with the highest rates observed in the upper 6 m below the SWI.

**Sedimentary iron phases.** Lake Towuti's sediment has extremely high total extractable Fe concentrations (Fig. 2), with maximum values >2500 μmol Fe cm$^{-3}$ (20% dry wt.). However, in most sediments studied to date, only a fraction of the total Fe pool is geochemically and biologically reactive and has the capacity to participate in redox-reactions associated with organic matter

mineralization and sediment diagenesis[21]. We thus conducted a suite of selective sequential extractions to determine both the reactive fraction of the sedimentary Fe pool and its transformations during organic matter mineralization and diagenesis in Lake Towuti's sediments (Fig. 2).

Extraction with 0.5 N HCl ($Fe_{HCl}$) captures the non-crystalline to poorly crystalline ferric iron (Fe(III) (oxyhydr)oxides, e.g., ferrihydrite, lepidocrocite), generally considered to be the most available to Fe-respiring microorganisms[22], as well as corresponding respiration products including sorbed Fe(II), poorly crystalline siderite, and green rust[23,24]. The $Fe_{HCl}$ fraction comprises up to 30% of the total extractable Fe pool. Within the $Fe_{HCl}$ fraction, Fe(II) is abundant, while Fe(III) is below our limit of detection (10 μmol cm$^{-3}$) at all depths. So, despite very high total Fe concentrations, Fe(III) phases considered readily available to Fe-respiring microorganisms[22] are virtually absent from Lake Towuti's sediments. This underscores previous observations that highly biologically reactive ferric iron minerals like ferrihydrite or ferrihydrite-like phases are entirely exhausted through respiration in the water column and very uppermost sediment layer[25]. We also targeted more crystalline carbonate phases like siderite with a sodium acetate ($Fe_{aca}$) extraction, which liberated an appreciable amount of Fe(II) but no detectable Fe(III). Concomitantly high DIC and Fe(II) concentrations translate

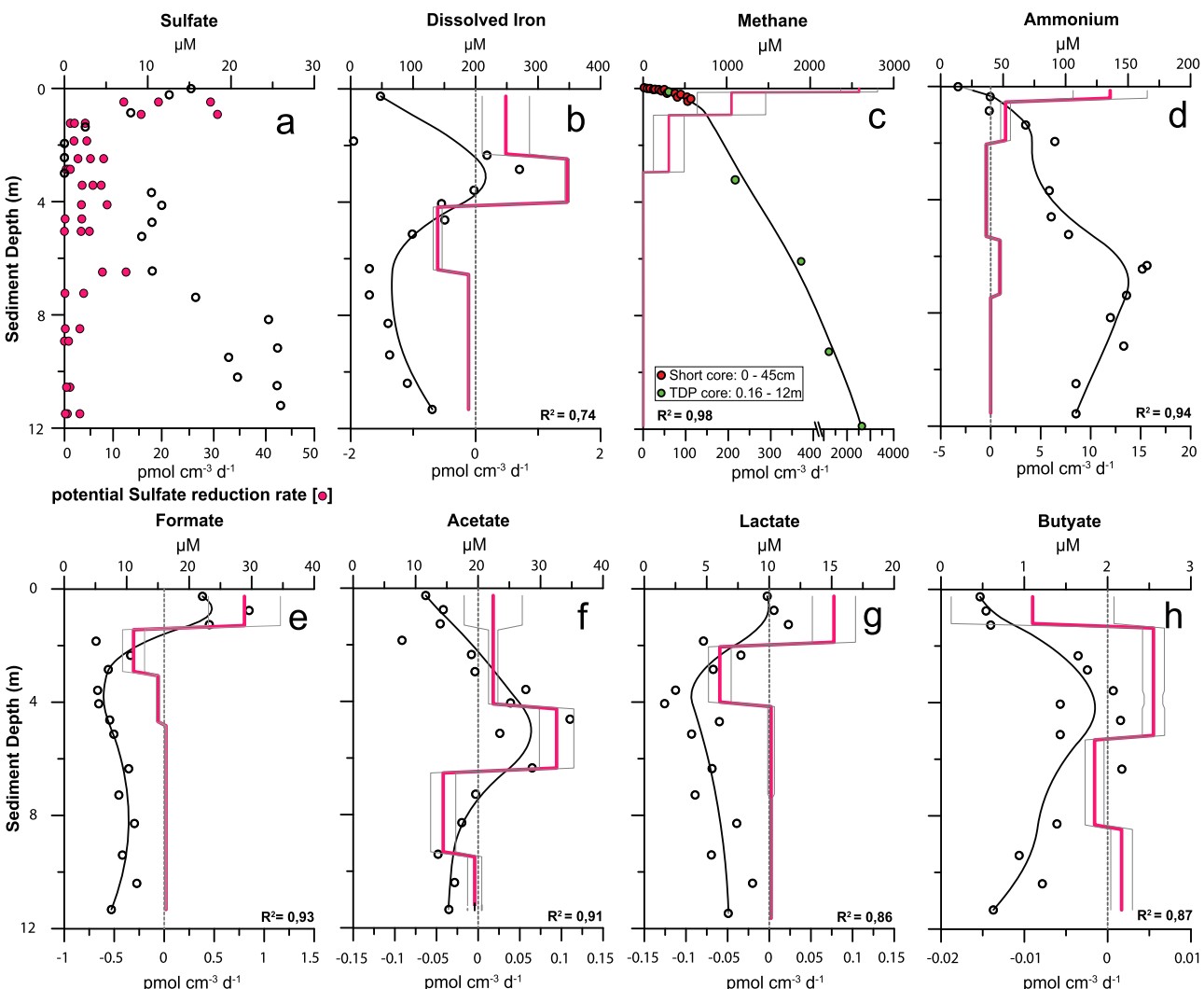

**Fig. 3 Pore water chemistry and turnover of reactants involved in organic matter mineralization.** Open circles show concentrations in sediment pore water. The black lines show the modeled concentration profiles, $R^2$ showing the correlation coefficient of the fit. The pink solid line shows the calculated mean reaction rate profile together with its standard deviation (gray line). Top x-axis shows the concentration, bottom x-axis indicates the reaction rate, positive values indicate net production, negative values net consumption of the respective compound. Error bars were omitted for clarity; reproducibility of the measured concentrations is always better by 5%. For the potential sulfate reduction rates (pSRR), measured via radiotracer incubations (pink circles in panel **a**), the individual measurements are plotted. Rates of potential sulfate reduction (**a**) and iron reduction (**b**) are much smaller than rates of methanogenesis (**c**). Net production of ammonium (**d**) and various volatile fatty acids (**e–h**) indicates that there is sufficient sedimentary organic matter to fuel heterotrophic microbial activity. See Supplementary Information for details.

to pore waters that are supersaturated with respect to siderite (FeCO₃) throughout the sediment (Supplementary Table S3)[26]. Despite the high abundance of acetate extractable Fe(II) and pore water supersaturation, siderite only accumulates in sporadic carbonate-rich layers[26], and mineral carbon is low throughout most of the sediment[27]. Much of this acetate extractable Fe(II) and by extension mineral carbon and siderite likely forms in response to Fe(II) reduction in the water column[25], as previously shown, though we do not rule out the possibility for transient diagenetic carbonate mineral formation.

Fe(III) phases not extracted in 0.5 N HCl, like goethite, hematite, magnetite, and some Fe(III)-bearing clays can also be respired under some laboratory and environmental conditions[28], and such phases have been shown to react with hydrogen sulfide in marine sediments[29]. These phases are thus also considered part of the reactive Fe(III) pool and expected to play a role in sediment diagenesis. We therefore targeted these phases using reductive sodium dithionite extractions (Fe$_{dith}$)[23] and found them to be

abundant in Lake Towuti's sediments, comprising up to 880 μmol cm⁻³ or 42% of the total Fe pool (Fig. 2). The mineralogical composition of dithionite extracted Fe(III) was further evaluated using quantitative X-ray diffraction, which identified the main Fe (III)-bearing phase throughout the core as goethite (Fig. S4 and Table S6). Reactive Fe present in magnetite was targeted using an ammonium oxalate/oxalic acid leach (Fe$_{oxa}$). Together, Fe$_{dith}$ and Fe$_{oxa}$ represent a theoretically bioavailable Fe(III) pool which accounts on average 320 μmol cm⁻³ or 31% of the total Fe, but this pool shows little variation with depth. The only exception is a notable decrease in dithionite extractable Fe concentrations in the upper 1 cm below the SWI (Fig. 2).

**Modeling of iron reduction.** The apparent lack of Fe(III) reduction in much of the sediment is consistent with the pore water profiles of Fe²⁺ concentration (Fig. 3) and pH (Fig. 4), the latter would be expected to increase in response to appreciable Fe(III)

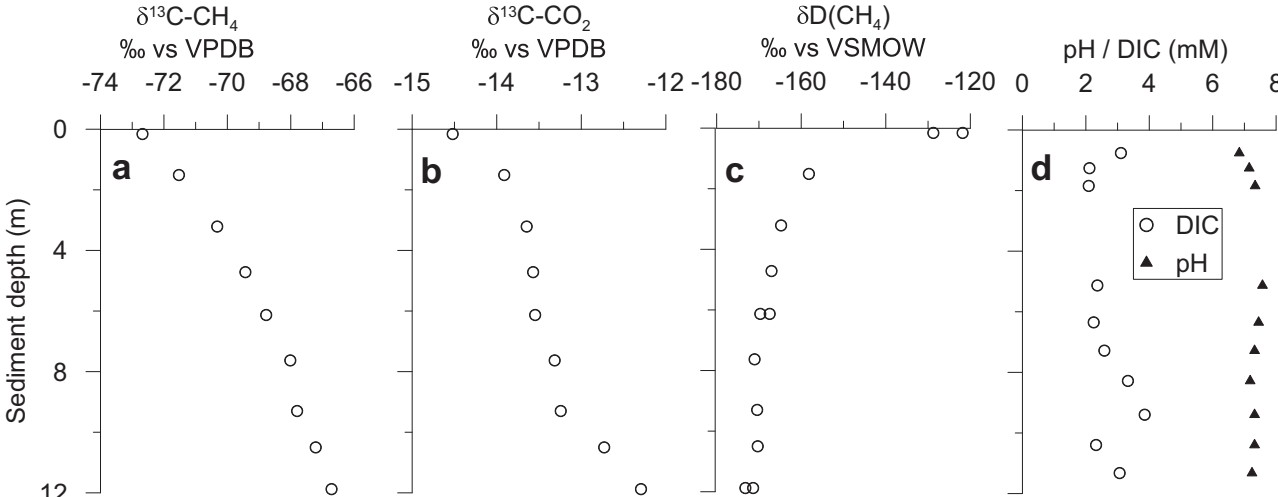

**Fig. 4 Stable isotopic composition of methane and CO₂ dissolved in Lake Towuti sediment pore water as well as pore water's pH and dissolved inorganic carbon concentration. a** Stable carbon isotopic composition of methane. **b** Stable carbon isotopic composition of carbon dioxide. **c** Deuterium composition of methane. **d** Pore water pH and DIC concentration. Each point represents an individual measurement. See Supplementary Information for methodological details.

reduction as this is a proton-consuming process[4]. Modeling based on diffusive fluxes of pore water $Fe^{2+}$ indicates very low net rates of background Fe(III) reductive dissolution (~1 mmol m$^{-2}$ yr$^{-1}$) in the upper 4 m (Fig. 3). Modeling based on solid phases indicates that Fe(III) reduction rates are highest in the upper 1 cm, just below the SWI (53 mmol m$^{-2}$ yr$^{-1}$), which compares to the rate of 30 mmol m$^{-2}$ yr$^{-1}$ observed in Lake Towuti's water column[25], and the depth-integrated Fe(III) reduction rate for the remaining 12 m is 105 mmol m$^{-2}$ yr$^{-1}$, resulting in a total depth-integrated Fe reduction rate of 160 mmol m$^{-2}$ yr$^{-1}$ over the upper 12 m. Assuming a 4:1 stoichiometry of iron reduction coupled to organic carbon oxidation[30], this translates to an organic carbon degradation rate of 40 mmol m$^{-2}$ yr$^{-1}$, which is low compared to total organic carbon accumulation rates at the SWI. These observations thus reveal that a large fraction of the microbial Fe reduction in Lake Towuti is restricted to the water column[25] and uppermost sediment layer. At the same time, this also reveals that the Fe(III) phases found throughout the sediment studied are stable for over tens of thousands of years.

**Quantification of sulfate reduction rates.** Given the apparently minor role of Fe reduction in light of the relatively high abundances of canonically reactive Fe(III) and labile organic matter, we explored other pathways of organic matter mineralization. Sulfate reduction commonly follows iron reduction in order of decreasing free energy yield in marine sediments[4]. Pore water sulfate concentrations in Lake Towuti are extremely low and decrease from 15 μM at the SWI to below our detection limit (1 μM) in the upper cm of the sediment (Fig. 3). Nevertheless, while geochemical modeling only predicts appreciable sulfate reduction in the upper 4 m (Supplementary Fig. S2), radiotracer incubation experiments reveal potential for sulfate reduction over the entire 12 m depth interval (Fig. 3). Depth-integrated rates of measured potential sulfate reduction (pSRR) are 20 ± 10 mmol m$^{-2}$ yr$^{-1}$, while modeled depth-integrated rates of sulfate reduction are 0.2 ± 0.15 mmol m$^{-2}$ yr$^{-1}$, which correspond to an organic carbon oxidation rate of 40 ± 20 and 0.40 ± 0.3 mmol m$^{-2}$ yr$^{-1}$, respectively, based on a 1:2 stoichiometry between sulfate reduction and organic carbon oxidation. Given that the measured pSRR rates intrinsically overestimate in situ rates and should thus be taken as an indication of metabolic potential only (see Supplementary

Information for details). We thus assumed that the measured pSRR and the modeled rates represent the respective upper and lower estimates of true sulfate reduction rates, respectively. We conclude that, like iron reduction, sulfate reduction plays only a minor role in organic matter degradation. Nevertheless, and importantly, the observation that sulfate reduction persists throughout the core confirms microbial reactivity of organic matter in these sediments.

**Quantification of methanogenesis rates.** Methanogenesis is commonly considered to be the final step in organic matter mineralization as it has the lowest free energy yield in the canonical cascade of early diagenetic redox reactions. In Lake Towuti, pore water methane concentrations increase continuously from 23 μM at the SWI to 2600 μM at 12 m depth (Fig. 3). The accumulation of methane throughout the sediment and the concave-upwards shaped concentration profile imply that methanogenesis occurs throughout the sediment. This is further supported by our modeling based on pore water methane concentrations (Fig. 3 and Supplementary Fig. S3) as well as by incubation-based measurements of potential hydrogenotrophic and acetoclastic methane production (Supplementary Fig. S1). Sediment samples from three depths (0.36, 2, and 7.4 m), reveal the potential for methanogenesis at these depths and by both pathways. The fact that methane concentrations at the SWI are not zero indicates a diffusive methane efflux out of the sediment into the water column (Fig. 3 and Supplementary Fig. S3). Our modeling results are also supported by the stable isotopic composition of pore water $CH_4$ and $CO_2$. The $\delta^{13}C$- and $\delta D$-values of methane are around −70‰ vs. VPDB and −170‰ vs. VSMOW, respectively (Fig. 4). Despite some variation with depth these values are all in the range observed for biogenic methane. The almost parallel increase of $\delta^{13}C(CH_4)$ and $\delta^{13}C(CO_2)$ is consistent with the consumption of $CO_2$ by hydrogenotrophic methanogenesis[31]. The modeled depth-integrated rate of methane production over the upper 12 m was 220 ± 90 mmol m$^{-2}$ yr$^{-1}$ (Fig. 3, Supplementary Table S4). Assuming a 2:1 stoichiometry for the conversion of organic matter to methane, methanogenesis accounts for the conversion of 440 ± 170 mmol m$^{-2}$ yr$^{-1}$ organic carbon. This rate far exceeds those of all other carbon mineralization processes combined but is still less than, and therefore

well supported by, the carbon accumulation rate. Within the upper 12 m of sediment, sulfate reduction, iron reduction, as well as methanogenesis combined add up to a total organic carbon mineralization rate of $519 \pm 191$ mmol m$^{-2}$ yr$^{-1}$, with methanogenesis being the dominant process (85–92%) followed by iron reduction (8%) and sulfate reduction (<1–7%).

## Discussion

Our data demonstrate that methanogenesis is the dominant (>85%) pathway for carbon mineralization in Lake Towuti sediment. The pSRR rates suggest that sulfate reduction may occur throughout the upper 12 m of sediment, but here it makes a minor, even insignificant, contribution to total organic carbon mineralization. Fe-reduction, likewise, is active in the water column and uppermost sediments of Lake Towuti, where it drives formation of primary authigenic magnetite[25] and other Fe(II) containing minerals[24], but remarkably it plays a minor role in the sediments despite the high abundance of Fe(III)-containing minerals. This observation emphasizes the role of primary water column processes in controlling sediment mineralogy in ferruginous environments and implies that the mineral products of Fe-respiration have strong potential to record primary water column signals. The apparent stability of Fe-oxides in Lake Towuti contrasts with the expected reactivity towards biological Fe(III) reduction based on both laboratory and environmental experiments with nanocrystalline to crystalline goethite and hematite[10,28]. This apparent lack of reactivity may be linked to the ultimate source of Lake Towuti's detrital iron (oxyhydr)oxides from the surrounding soils, which can be highly crystalline with low surface areas[32]. Prior studies at Lake Towuti, however, suggest that regardless of source, sedimentary iron (oxyhydr)oxides are poorly crystalline and may have experienced strong reworking prior to burial[16]—such conversion to authigenic phases would likely render them more reactive towards reduction. Perhaps a more likely explanation for low reactivity is surface passivation through Fe(II) sorption[33,34], which would be important at the 10–100 s of μM Fe(II) present in Lake Towuti's pore waters (Fig. 3).

Microbial methanogenesis produces most of the methane on Earth, but in modern marine sediments with abundant sulfate AOM consumes more than 90% of the total methane produced, thereby providing a buffer between sedimentary methane production and the atmosphere[5]. In environments where sulfate is scarce, AOM has been linked to the reduction of nitrate[35,36] and Fe(III)[37]. With nitrate and nitrite below our limit of detection (4 μM) and very little sulfate, Fe-dependent AOM remains the only pathway with potential to quantitatively consume methane in Lake Towuti's sediment. While it is true that tightly coupled methane production and oxidation could be masked in pore water profiles[38], the process would cause a decline in oxidant concentration with increasing depth, in this case Fe(III), which we do not observe. Pore water profiles also show no evidence for net methane consumption and Fe-reduction rates are small, so we thus conclude that AOM is generally negligible in Lake Towuti's sediments, as the known electron acceptors are either not available (NO$_3$, NO$_2$, SO$_4$) or not utilized (Fe(III)).

Previous studies have shown the metabolic potential for coupled Fe, S, and C cycling[39], which could possibly operate cryptically[40]. Such cryptic cycles are important in many environments and we cannot rule them out in Lake Towuti's sediments. However, even cryptic cycles are constrained by mass balance. In the case of Lake Towuti, a cryptic sulfur cycle would need to be supported by an oxidant and the only oxidant of sufficient abundance to sustain appreciable cryptic sulfur cycling throughout the core is Fe(III). Since Fe(III) is largely stable downcore, mass balance dictates that cryptic sulfur cycling would

have to operate at rates much lower than Fe(III) deposition and burial and thus, we argue that such cryptic sulfur cycling is insignificant to overall net sediment Fe, C, and CH$_4$ budgets.

Though the specific reasons for the apparent lack of Fe(III) reactivity in Lake Towuti remain somewhat uncertain, the dominance of methanogenesis has strong implications for coupled carbon and iron cycling in the Precambrian oceans. Robust extension of our results to these Eons, however, depends on comparisons between the authigenic and detrital Fe mineral phases in Lake Towuti and those of mainly hydrothermal provenance expected to form in the Precambrian oceans. Notably, crystallinity, through its control on surface area of the relevant phases, would be expected to play an important role in dictating reactivity[41]. While poorly crystalline goethite dominates Lake Towuti's Fe pool throughout the sediment studied (Supplementary Fig. S4), Fe (oxyhydr)oxides in the Precambrian oceans may have had even greater surface areas and therefore reactivity[28,41]. There is considerable debate on the nature of the primary Fe phases deposited from the Precambrian oceans. High, but uncertain, concentrations of seawater silica would likely have hindered the transformation of poorly crystalline hydrous ferric oxides into more crystalline, and therefore less reactive phases like the poorly crystalline goethite found in Lake Towuti[42]. We note, however, that the silica concentrations in Lake Towuti and catchment waters (~300 μM) indeed approach the lower end of experimentally estimated silica concentrations (600–1500 μM) in the Precambrian oceans[43] and we thus speculate similar, but possibly muted, effects on Fe speciation in Lake Towuti. Nevertheless, given that the Fe (oxyhydr)oxides in Lake Towuti may be less biologically reducible than those perceived for the Precambrian oceans, we take our results from Lake Towuti as an end member scenario for reconstructing the role of Fe reduction in Precambrian marine organic matter mineralization. In contrast, we consider prior work from Chocolate Pots hot springs, where silica concentrations reach 2.5 mM[11], as the possible other end member. Such high silica concentrations, which are higher than what has been estimated for the Precambrian oceans, lead to preservation of poorly crystalline, silica-rich hydrous ferric oxides[11], of which ~80% of their Fe(III) can be reduced through organic matter oxidation[11,12].

Photosynthesis in the Archean and Proterozoic Eons would have led to the production of organic matter, and under ferruginous ocean conditions, the deposition of Fe(III)-rich sediments, either through direct photosynthetic Fe-oxidation (photoferrotrophy)[15] or through reaction of Fe(II) with oxygen produced through oxygenic photosynthesis[44]. Rates of organic matter production through photosynthesis during much of the Precambrian Eons were likely between 1% and 10% of modern values, or 40–400 Tmol yr$^{-1}$[45,46] and if organic carbon burial efficiency was comparable to the modern (1–10%)[47,48], this translates to global mineralization rates between 36 and 396 Tmol yr$^{-1}$. While both Fe-reduction and methanogenesis likely evolved early in the Archean Eon[49,50], analogy to Lake Towuti would suggest that much of the Fe(III) produced through photosynthesis could have been buried without appreciable respiration and reduction coupled to oxidation of co-deposited organic carbon. In contrast, if silica-rich hydrous ferric oxides, like those from Chocolate Pots are closer in reactivity to the Fe(III) produced through photosynthesis in the Archean Eon, then a greater fraction of the photosynthetic biomass would have been respired through Fe(III) reduction. These analogies can be tethered to the geological record by considering the redox state of Fe preserved in BIFs, which on average is 2.6[51] (or 2.4 based on other sources[52], with possible variation from 2.0 for a pure siderite end member, to 3.0 for a hematite end member), which implies that at least 60% of the Fe(III) is commonly preserved and buried. While this

preservation can, in part, be explained through physical separation of Fe(III) from organic matter[51], it also likely reflects the reactivity of Fe(III) phases. Variability (2.08–2.97) in redox state across facies, between deposits, and through time could also thus be partly explained, and even provide evidence for Fe(III) reactivity as a controlling factor in the extent of microbial Fe(III) reduction. The extent of decoupling from Fe-reduction would have controlled organic matter accumulation in sediments and rates of methanogenesis.

To assess potential scenarios for carbon mineralization via methanogenesis under different Fe(III) reactivities we used Lake Towuti's sediment as the low Fe(III) reactivity end member, and silica-rich, hydrous ferric oxide sediment like Chocolate Pots[11,12], as the high Fe(III) reactivity end member. For the low Fe(III) reactivity end member, breakdown of 85–92% of organic carbon through methanogenesis would have led to marine $CH_4$ production of 15–180 Tmol $yr^{-1}$. For comparison, analogy to Chocolate Pots, where ~80% of the Fe(III) is reduced[11,12], would have led to marine $CH_4$ production of 4–40 Tmol $yr^{-1}$. Such estimates are in fact supported by the redox state of Fe in BIFs, which are often mixed-valence, both demonstrating preservation of Fe(III) and indicating either partial reduction of deposited Fe(III) minerals or the deposition of primary Fe(II)-bearing phases. Mineralogical and isotopic observations, in fact, are increasingly pointing towards primary deposition of Fe(II) minerals with a comparatively minor role for diagenetic Fe(III) reduction[53,54], which is in line with the primary locus of Fe(II) reduction and mineral formation in Lake Towuti's water column, and strongly implicates primary processes in setting the mineralogy of BIFs and other ferruginous sediments.

In oceans lacking appreciable oxygen and sulfate[7], methane produced in sediments would, in the absence of AOM, have led to equivalent effluxes of $CH_4$ to the overlying anoxic oceans and atmosphere. Atmospheric $CH_4$ concentrations can be approximated through new solutions to photochemical models for a given biospheric $CH_4$ production flux[55], which imply a wide range of concentrations from 40 up to 24,000 ppmv, depending on biological productivity, organic carbon burial efficiencies, and importantly, the reactivity of sedimentary Fe minerals towards biological reduction. Such strong upper predicted Precambrian biospheric $CH_4$ fluxes and atmospheric $CH_4$ concentrations, versus previous work[55], are largely the result of the high conversion efficiency of organic carbon to $CH_4$, as observed in lake Towuti's sediments. We note that atmospheric methane concentrations in the Precambrian may thus be a function of Fe(III) mineral crystallinity and reactivity and could have varied strongly in response to changes in the factors that control this reactivity, including seawater silica concentrations. Resolving the wide range in possible Precambrian atmospheric methane concentrations thus requires both tighter constraints on Precambrian primary production, as well as the nature of the primary Fe(III) mineral precipitates and their dynamics through time. However, irrespective of the reactivity of Fe(III) minerals, methanogenesis is likely a quantitatively important pathway for carbon mineralization in ferruginous sediments.

## Methods

**ICDP drilling campaign and sampling procedure.** In 2015 a scientific drilling campaign took place on Lake Towuti within the framework of the International Continental Scientific Drilling Program (ICDP). We retrieved a ~115 m long sediment core dedicated for geomicrobiological investigations at Drill Site 1 at a water depth of 153 m using the ICDP Deep Lakes Drilling System[17]. We used a tracer to monitor infiltration of drilling fluid into the core[56] and only used uncontaminated samples for our analyses. Our study focused on the upper 12 m of the drill core, a section equivalent to sediments that have already been subject to comprehensive paleoclimatic investigations, which also supplied sediment ages and estimates of sedimentation rates[18]. Cores were collected in HQ-size butyrate liners

(66 mm core diameter) in 3 m intervals using hydraulic piston coring. After retrieval, sediment cores were cut into two subsections of 1.5 m length. In addition, short (<0.4 m) sediment cores were retrieved from the same site using a small gravity-coring device that recovered an undisturbed SWI and allowed interrogation of the uppermost sediments in more detail.

**Pore water sampling and analysis.** Pore water was squeezed under anaerobic conditions. Concentrations of major cations and anions were analyzed by ion chromatography. Dissolved iron and phosphate concentrations were determined spectrophotometrically[57,58]. Concentrations of VFAs in the pore water were measured by 2-dimensional ion chromatography mass spectrometry (2D IC-MS)[59].

**Iron speciation.** For iron speciation, a subsample of 500 mg of wet sediment from each core interval of both sediment cores was extracted in the field and immediately leached in 1 mL 0.5 N HCl, and Fe-speciation (Fe(II) and Fe(III)) of the easily extractable Fe-phases was measured spectrophotometrically on site using a ferrozine assay[57,60]. The complete Fe-speciation protocol was performed on anoxically preserved and freeze-dried sediments milled to fine powders using an agate hand mortar and pestle. Sample masses of 200 mg of sediment were weighed into 15 mL centrifuge tubes, and subjected to the Fe-speciation sequential extraction scheme based on the protocol of Poulton and Canfield[23]:

- Leaching the sample with 0.5 N HCl for 1 h extracted the highly reactive Fe in hydrous ferric oxides like ferrihydrite or lepidocrocite ($Fe_{HCl}$). Iron speciation was determined spectrophotometrically[57,60]. However, easily reducible ferric iron within the $Fe_{aca}$ fraction was always below the limit of detection in all samples, so that this fraction entirely consists of ferrous iron.
- Fe(II) in carbonate and poorly crystalline phases was extracted by leaching the sediment with a sodium acetate solution adjusted with acetic acid to pH 4.5 with acetic acid for 48 h ($Fe_{aca}$).
- Reactive Fe(III) in goethite and hematite was extracted by treating the sediment sample with a sodium dithionite solution (50 g $L^{-1}$) buffered to pH 4.8 with 0.35 M acetic acid/0.2 M sodium citrate for 2 h ($Fe_{dith}$). Due to the reductive dissolution in this step, the resulting extract only contains Fe(II). However, as the main minerals dissolved in this step only contain Fe(III) we consider this pool to consist only of Fe(III). We further verified the selectivity of the extraction by performing a dithionite leaching experiment on two samples from the upper 30 cm of the Towuti sediment. We first quantified via qXRD the amount of goethite in these samples, and then subsequently leached them with dithionite. Our results (Table S6) demonstrate that within error of the qXRD and Fe-speciation measurements, dithionite quantitatively extracts goethite in the Lake Towuti sediments. Furthermore, XRD data demonstrates the presence of goethite throughout the core (Supplementary Fig. S4), and thus we conclude that Fe contained in our dithionite extractions was indeed goethite.
- Leaching the sediment sample with 0.2 M ammonium oxalate/0.17 M oxalic acid solution (pH 3.2) for 6 h extracted the reactive Fe present in magnetite ($Fe_{oxa}$). Given the mixed valence of Fe in magnetite, 2/3 of the extracted Fe was counted as Fe(III) and 1/3 as Fe(II).
- Finally, samples were subjected to a near boiling 6 N HCl extraction for 24 h to extract the remaining unreactive Fe in silicates and Fe-bearing clays like nontronite ($Fe_{Sil}$). Fe(II)/Fe(III) ratios were not determined.

Our reactive Fe pool is defined as carbonate-associated Fe ($Fe_{aca}$, sodium acetate extractable Fe), hydrous ferric oxides ($Fe_{HCl}$, 0.5 N HCl extractable Fe), ferric (oxy) hydroxides ($Fe_{dith}$, dithionite extractable Fe), and magnetite ($Fe_{oxa}$, oxalate extractable Fe), (Reactive Fe pool = $Fe_{aca}$ + $Fe_{HCl}$ + $Fe_{dith}$ + $Fe_{oxa}$). The total reactive ferric Fe pool is defined as ferric iron present within the $Fe_{HCl}$, $Fe_{dith}$, and within the $Fe_{oxa}$ fraction. The total reactive ferrous iron pool is composed of the ferrous iron within the $Fe_{aca}$, $Fe_{HCl}$, and $Fe_{oxa}$ fraction. Finally, the total iron pool is defined as the sum of all reactive Fe phases and non-reactive (lithogenic) Fe contained in silicate minerals ($Fe_{Sil}$).

Our extractions dissolved >95% of the Fe from the PACS-2 international reference standard. All Fe concentration measurements were performed using a flame atomic absorption spectrophotometer (flame AAS). Precision on triplicate measurements was 1.2% and our limit of detection was 1500 μg $g^{-1}$ (0.15 wt% or ~10 μmol $cm^{-3}$).

**TOC analysis.** Total organic carbon was quantified by Rock-Eval 6 pyrolysis (Vinci Technologies).

**Methane concentrations and isotopic analysis.** To minimize losses due to outgassing, sediment samples for methane concentration and isotopic analysis were taken with a cutoff syringe immediately after retrieval of the core and stored in glass vials filled with saturated NaCl solution without headspace. At least 24 h prior to analysis we introduced 3 mL of Helium as headspace. Methane concentrations were quantified by gas chromatography, isotopic composition measured with a continuous-flow isotope ratio mass spectrometer. See Supplementary Information for analytical details.

**pSRR rates**. pSRR rates were determined by incubation with radioactive $^{35}SO_4^{2-}$[61] using sterile glass plugs fitted with a syringe plunger to obtain undisturbed sediment mini-cores, the end was closed with butyl rubber stoppers. Due to legal constraints it was not possible to carry out the radiotracer incubations on site. We therefore collected WRC, stored them in an $N_2$ atmosphere and retrieved the subsamples for radiotracer incubations from the WRC several weeks after the drilling back in the home lab in Potsdam. After pre-incubation for 24 h at the approximately in-situ temperature of 30 °C, ca. 100 kBq of $^{35}SO_4^{2-}$ tracer, containing ~10 μM of non-radioactive $SO_4^{2-}$ in order to avoid complete turnover of the sulfate pool[62] was injected into each sample. Samples were incubated for 24 h at 30 °C in the dark. Incubations were stopped by transferring the samples into 10 mL of 20% zinc acetate solution. The microbially reduced inorganic sulfur species were separated from the remaining sample and the unreacted sulfate tracer using the cold chromium distillation[63]. Radioactivity was quantified by liquid scintillation counting. Because of the time lag between recovery of the core and the start of incubations the in-situ sulfate concentration and the autochthonous microbial community might have changed. We also added non-radioactive sulfate to the tracer. As we cannot quantify how these factors affect the sulfate reduction rates, we consider them to be potential, as the only conclusion that can be drawn unequivocally is that there is a microbial community that is able to perform sulfate reduction.

**Geochemical modeling**. Net reaction rates of dissolved chemical species were calculated using the MATLAB script of Wang et al. (2008)[64], assuming that the pore water concentration profiles represent steady-state conditions.

**Potential methane production**. The potential for biogenic methane production was investigated by incubation experiments with sediment samples from three different depths (0.36, 1.95 and 7.4 m). The sediment was slurried using sulfate-depleted freshwater medium mimicking the pore water concentrations of Lake Towuti sediment (Supplementary Table S2). For incubations for hydrogenotrophic methanogenesis, the butyl stoppered glass crimp vial was flushed with a mixture of $H_2/CO_2$ (80/20%).

## Data availability
Pore water geochemistry and bulk sediment measurements of downcore profiles from site TDP-1A of the ICDP Towuti Drilling Project, Lake Towuti, Indonesia. https://doi.org/10.1594/PANGAEA.908080. All other data discussed in the paper will be made available to readers in the supplement.

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

## Acknowledgements
This research was carried out with partial support from the International Continental Scientific Drilling Program (ICDP), the U.S. National Science Foundation (NSF), the German Research Foundation (DFG), the Swiss National Science Foundation (SNSF), PT Vale Indonesia, the Ministry of Research, Education, and Higher Technology of Indonesia (RISTEK), GFZ German Research Centre for Geosciences, the Natural Sciences and Engineering Research Council of Canada (NSERC), and Genome British Columbia. We thank PT Vale Indonesia, the US Continental Scientific Drilling and Coordination Office, and US National Lacustrine Core Repository, and DOSECC Exploration Services for logistical support. The research was carried out with permissions from RISTEK, the Ministry of Trade of the Republic of Indonesia, the Natural Resources Conservation Center (BKSDA), and the Government of Luwu Timur of Sulawesi. We thank Tri Widiyanto and his staff from Research Center for Limnology, Indonesian Institute of Sciences (LIPI) for their administrative support in obtaining the Scientific Research Permit. Special thanks are due to Jenny Wendt and Xavier Prieto for their assistance during methane analysis. Jan Axel Kitte is acknowledged for his support in the field and in the laboratory.

## Author contributions
A.F., S.A.C., J.K. designed the study and wrote the manuscript; A.F., K.B., C.G., L.O., V.B.H., A.V., S.N., R.S. performed experiments; H.V., J.M.R., C.H., S.B. organized and managed the field campaign; H.V., J.M.R., D.A., D.W., M.M. contributed to the writing of the manuscript.

## Funding

## Competing interests
The authors declare no competing interests
