## [Peer Review File · Nature Communications]

REVIEWER COMMENTS

Reviewer #1 (Remarks to the Author):

This paper documents the biogeochemistry of Fe-rich, S-poor, permanently anoxic sediments of Lake Towuti in Indonesia, with the goal of evaluating the importance of microbial Fe(III) oxide reduction versus methanogenesis in sediment carbon metabolism. Previous work in analogous, nearby Lake Matano concluded based on water column methane profiles and mass balance calculations that methanogenesis is likely to be the dominant anaerobic carbon oxidation pathway (Crowe et al., *Geobiology*, 2011). However, to date no studies have reported a detailed analysis of Fe geochemistry in the sediments of any of the ferruginous lakes in Indonesia, i.e. measurements that could constrain more directly the potential contribution of Fe(III) reduction to sediment carbon metabolism. Thus, the paper is timely and advances our understanding of the biogeochemistry of these unusual modern ecosystems.

More broadly, the ferruginous Indonesian lakes provide insight into biogeochemical processes in Precambrian marine systems, i.e. prior to and after the Great Oxidation Event when Earth's oceans were likely rich in Fe(II) (and low in sulfate), and when oxidation of Fe(II) in surface waters likely led to the delivery of large quantities of Fe(III) oxides to surface sediments, where they could have served as electron acceptors for sediment carbon oxidation. In fact, the main underlying goal of this paper is to use the results from Lake Towuti to place constraints on the extent to which Fe(III) oxide reduction may or may not have significantly influenced the production and release of methane from Precambrian marine sediments. An enormous body of literature has been devoted to understanding the signatures of microbial Fe(III) oxide reduction in Precambrian marine sediments (e.g. ones that led to Banded Iron Formations), and this paper seeks to expand to our understanding of the potential quantitative role of this pathway, using the Fe-rich sediments of Lake Towuti as a modern analog. In fact, there is a real paucity of such modern analogs, owing to the high abundance of sulfate in modern oceans which ultimately causes the sulfur cycle (i.e. sulfate reduction) to impinge on sediment Fe cycling even in situations where Fe is relatively abundant.

The authors present a comprehensive set of biogeochemical measurements across the upper 10 m of sediment, including: dissolved (porewater) Fe(II); sequential selective extractions to speciate solid-phase Fe; sediment sulfate distribution and radiotracer sulfate reduction rate measurements; dissolved methane concentrations; $\delta^{13}\text{C}$ values for dissolved methane and dissolved inorganic carbon; and concentrations of short-chain organic acids that serve as electron donors for anaerobic microbial respiration. To summarize, both mass balance calculations and sediment geochemical modeling suggest that methanogenesis is in fact that dominant pathway for organic carbon metabolism in these permanently anoxic sediments. Although "reactive" Fe(III) oxide phases (i.e. those released via extraction with dilute HCl, ammonium oxalate, and/or citrate/sodium dithionite) are present in relatively high abundance (hundreds of $\mu\text{mol cm}^{-3}$), they show no systematic decline with depth below the top 1 cm of sediment. In addition, sulfate concentrations are near background throughout the sediment column, such that the potential contribution of sulfate reduction to carbon metabolism is minimal. Hence, as inferred previously for Lake Matano, this study demonstrates explicitly and convincingly that the vast majority (ca. 90%) of sediment carbon metabolism is mediated by methanogenesis in the ferruginous lakes of Indonesia.

The results from the ferruginous Indonesian lakes have important implications for sediment carbon metabolism in what are assumed to be ferruginous Precambrian marine systems. However, a key issue regarding the use of the Indonesian lakes as direct analogs to Precambrian marine sediments relates to nature and origin of the presumably "reactive" – but apparently non-microbially-reducible – Fe(III) oxide phases that persist in these Fe-rich sediments. In the introduction to the paper the authors state that "Intensive weathering of ophiolitic bedrock from the catchment supplies the lake with a strong influx of iron(oxy)hydroxides". The problem here is that this source of Fe input is fundamentally different from the source of Fe(III) oxide that was likely delivered to Precambrian marine surface sediments. In the case of the latter, it was almost certainly oxidation of Fe(II) in the water column (e.g. via either anoxygenic photosynthesis or reaction with oxygen produced by oxygenic photosynthesis), rather than inputs from continental weathering, that was

the dominant source of Fe(III) oxide input. Given the relatively high concentrations of dissolved silica in Precambrian oceans, it seems likely that the Fe(III) oxide phases delivered to the sediments were actually oxide-silica coprecipitates, which experimental studies have shown to be subject to near-complete reduction by dissimilatory Fe(III)-reducing bacteria (Percak-Dennett et al., *Geobiology*, 2013; Fortney et al., *Geobiology*, 2016). It seems possible that the large quantities of non-reducible, crystalline phases such as goethite and hematite represent mainly inputs from the surrounding catchment rather than oxidation of Fe(II) in the water column. In addition, even Fe(III) oxides produced via anoxygenic photosynthesis (evidence for which, notably, was provided by Crowe et al., *PNAS*, 2008 in Lake Matano) or reaction with oxygen in the photic zone could result in relatively rapid transformation to crystalline phases in the absence ligands such as phosphate (commonly present in high abundance in microbiological culturing experiments) or silica that are well known to retard Fe(III) oxide crystallization during Fe(II) oxidation.

The upshot of the above considerations is that the quantitative extension of the results from Lake Towuti and ferruginous Indonesian lakes to Precambrian oceans needs to be qualified with respect to the microbial reducibility of the Fe(III) oxide phases. To achieve this, the authors consider end-member scenarios where the Fe(III) oxides deposited to Precambrian marine sediments are assumed to have either high or low reducibility. The high reducibility scenario is based on results from Chocolate Pots hot spring in Yellowstone National Park, where Fe(III) oxide-silicate coprecipitates (formed where Si/Fe(II)-rich neutral-pH groundwater is discharged to the land surface) are known to be subject to extensive (ca. 80%) microbial reduction. The low reducibility scenario is based directly on the mass balance results from Lake Towuti sediments. In doing this, they are able to constrain the potential contribution of microbial Fe(III) oxide reduction versus methanogenesis to sediment carbon metabolism in Precambrian marine sediments with varying degrees of Fe(III) oxide reducibility. The results suggest that increased oxide reducibility, as might have been expected for Fe(III) oxide-silica coprecipitates formed in the water column, could have led to substantial decreases in the amount of methane released from Precambrian marine sediments. The authors incorporate a discussion of the average oxidation state of Fe oxide mineral phases in Precambrian sediments, which is useful for "tethering" their results to the geological record. The overall conclusion is that variations in Fe(III) oxide reducibility, and hence preserved Fe mineral oxidation state, could have varied across facies, between deposits, and through time, with attending potential impacts on sediment carbon metabolism and methane production.

In my opinion, this is a well-developed body of work that will be of broad interest to biogeoscientists interested in both modern and ancient ferruginous aquatic systems. The conclusion drawn for Lake Towuti (and by extension other analogous ferruginous Indonesian lakes) are well supported by the data, and the authors do a nice job of explaining things in a succinct, interesting manner. They also succeed in connecting the results from this modern environment to analogous Precambrian marine systems, thereby contributing to our evolving understanding of what controlled atmospheric methane concentrations before and after the Great Oxidation Event.

Reviewer #2 (Remarks to the Author):

It was a privilege to be asked to review the manuscript by Friese et al. entitled: "Iron oxide reactivity controls organic matter mineralization in ferruginous sediments", submitted for consideration to *Nature Communications*. First and foremost, my sincerest apologies must be conveyed to everyone concerned with this submission, for the delay in turning in my review. I fell ill recently and that set me back considerably, at a time when I am also overloaded with remote teaching responsibilities. The fact that I am based at a southern hemisphere university and thus find myself in the middle of the academic year, did not help the cause much either. I will elaborate a bit in my review, even though I never thought I would when I first received the manuscript. This is because I am recommending rejection of it, and my reasoning is based on a couple of truly fundamental shortcomings of this work which I cannot see how they can be adequately rectified through any kind of revision to the current version. In the following lines, I will try and provide an insightful enough justification for my recommendation, with the hope that the authors and the editors will see my concerns and my points clearly enough. It is important to

note that I read and assessed this work from my own specific viewpoint of someone who has worked on BIFs all their (scientific) lives. Therefore, my critique will revolve, in most part, around how the results and the arguments that the authors present, speak (or do not) truly and directly to the early Precambrian ferruginous ocean, and to the putative processes at play in that environment. After all, the discussion of the authors places its largest emphasis on precisely that aspect, given that lake Towuti and nearby lake Matano are variously considered as modern analogues of Precambrian ferruginous oceans and, by extension, of the iron-rich sediments that precipitate/d in them.

The first point I wish to raise might come across as a minor one, but is nonetheless strongly indicative of the direction that I am heading towards in this report: the statement in lines 68-70 about Fe-rich shale being the dominant sediment type in the early ferruginous ocean while BIF was an "extreme" end-member of iron deposition, is rather preposterous, if I may say so. I have read several times before the solitary citation accompanying that statement (i.e. Poulton & Canfield, 2007) and I did not quite see any support for such a contention; even if that was the case though, one wonders why the authors later in their Discussion, interpret their results exclusively in the context of the "extreme BIFs" and not of the "widespread ferruginous shales" they are referring to in their introductory lines.

Moving to the section "Sedimentary iron phases" (lines 138-167), the authors present geochemical results that are essentially the only proxy of the true mineralogy of the studied sediments.

Through standard sequential extraction, they have derived information about ferric iron availability in the sediment, that would constitute the "reactive iron pool" for organic carbon re-mineralisation. The result is quite astounding: the "easily" soluble fraction (i.e. that derived through acetate and 0.5M HCl digestions) returned no ferric iron whatsoever, and instead revealed the exclusive dominance of Fe(II), which makes up approximately one-third of the total iron pool! This, to me, means no ferrihydrite preservation whatsoever, unless (for some reason) the primary ferrihydrite did not react in weak HCl but only later during oxalate/dithionite extraction. This, however, is not at all clear, as there is no information presented by the authors that would constrain which phases exactly yielded ferric iron upon those two digestion steps. In fact, in the absence of any added information on the Fe(II)/Fe(III) ratio (I presume that the authors did titrate their sample solutions in order to be able to determine the oxidation state of Fe in the earlier extractions), one could conceivably argue that the ferric iron may have been dominated by magnetite dissolution, which means that Fe(II) might still be "lurking" here too. After all, magnetite dissolves (at least partially) in both oxalate and dithionite, and some of the authors of the current study have previously reported magnetite associated with the omnipresent siderite (Vuillemin et al, 2019). The truth, therefore, is this: if one were to dismiss the unreactive fraction of the sediments as representing recalcitrant, iron-bearing silicate detritus from the hinterland, then the actual budget of iron yielded through the speciation analyses is dominated by more than 50% Fe(II)! The very bulk of that iron ought to have been hosted in siderite, given both the current results and the recent study of Vuillemin et al. (2019) that deals with exactly that phase in the sediments and its possible origins. In view of the above, the question I am readily asking is this: how did that siderite exactly form? Unfortunately, the same paper (Vuillemin et al., 2019) does not provide a clear-cut pathway of siderite formation, and instead proposes a couple of possibilities, such as diagenetic replacement of carbonate green rust. (It should be noted here that siderite formation via calcic carbonate replacement during diagenesis has also been reported elsewhere: see Lin et al., *Geobiology*, 2019). If that is the case though, then one needs to really delve into the specifics of the ultimate source of Fe(II) in this stratified lake system: if the authors want to argue that there is methane generation in the sediment pile but no anaerobic oxidation thereof by ferric iron or sulphate, then something must be reducing iron, and that reduction must be efficient enough to lead to saturation in siderite! That saturation would have to be achieved either in the water column to produce primary siderite precipitates (a possibility also discussed by Vuillemin et al. 2019), or in the sediment, to justify replacement of other primary carbonate material by siderite.

These are all questions (and there are plenty more) that I cannot find any reasonable answers for in the current manuscript. Clearly, establishment of stratification and ferruginous conditions in a lake requires processes of iron redox cycling at play: if that hardly happens in the sediment column, it has to dominate in the water column itself. So, is it possible that bottom waters in lake Towuti are generating ferrous iron by reduction of primary ferrihydrite? If they do, what is the electron donor? Is it organic matter? In other words, is iron reduction as effective here as to happen already in the water column, in marked contrast to the arguments of the authors? I have to digress a bit here in order to add that current classic models for BIF diagenesis as put forward

by the Wisconsin and Tübingen groups over the years, invoke dissimilatory iron reduction as THE key pathway of organic matter cycling. The same models also predict that DIR would have operated exclusively below the sediment-water interface, and thus make no provision for it in the ambient water-column. Therefore, the BIF diagenetic models seem to be diametrically incompatible with what is interpreted by the authors to be happening in Lake Towuti, raising serious doubts as to how appropriate an analogue it is for ferruginous oceans in deep time. An alternative electron donor of course for water-column Fe cycling may have been the methane itself that the authors report: should one be led to assume then, that methane is not oxidized in the sediment but it may endure near-quantitative oxidation in the water column, thus leading to conditions favoring siderite saturation? The implications of such an interpretation are of course truly profound in the context of BIF, for more than one reasons: for example, coupled methane and iron cycling in the water column would result in the ongoing introduction of isotopically very light carbon into the DIC budget. This would definitely be inconsistent with the contention of a Neoproterozoic ocean containing high DIC abundances at near-0 per mil as proposed previously (e.g. see Fischer et al., *Precambrian Research*, 2009). The other issue, of course, is that if methane found itself into the contemporaneous atmosphere to act as potent greenhouse gas against a dim early sun (Thompson et al., *Science Advances*, 2019), one key reductant for iron redox cycling – and hence iron carbonate saturation – would be removed from the equation, leaving only organic matter available to “do the trick”.

I do not think there is any need for me to continue my thought process as conveyed in the foregoing paragraphs, simply because the possibilities of alternative interpretation are almost endless and the current manuscript fails to provide any meaningful constraints whatsoever towards resolving the various overlapping issues. The bottom line is this: on the one hand, the authors posit that diagenetic iron reduction is ineffective to the point where organic matter ends up turning into methane, which then oozes out of the sediment without suffering any measurable oxidation, and yet the sediment is full of siderite! There is a good deal that does not agree with me on this, as I have already argued in the preceding passages of text; another example would be the statement in line 343 that the average iron oxidation in BIF is 2.6, something that not only do I vehemently disagree with, but the abundant siderite in the Towuti sediments does too! Here is, in fact, where my biggest disappointment lies with this manuscript: if there is one proxy I could identify that has huge potential to provide at least some constraints, that is the carbon isotope composition of the siderite itself! To this day, I am completely baffled as to why the authors never did such analyses, or if they did, why they did not include the results here (although I can think of some reasons for the latter, but I would never insinuate that the authors would ever purposely withhold such important data). The carbon isotope composition of BIF siderites and other iron carbonate minerals (i.e. ankerite) is widely documented in literature and is well in the negatives, so it is almost a no-brainer that if one wants to constrain redox cycling of iron and carbon in a modern analogue to a BIF, the very first thing to analyze should be the carbon isotope composition of the contained iron carbonates. One must also not forget here that, unlike the Towuti sediments which contain percentage level organic carbon, classic BIF is practically organic matter-free globally, and that is precisely the feature exploited in the recent papers of Thompson et al (2019) and Dodd et al (2019), as some of the current authors would be able to attest to. It seems to me that the more one dives into the secrets of the Towuti lake, the less support they will find for its true analogy to BIF depositional systems.

I will conclude my report through a further reference to the methane-forming process the authors present in their paper. I have read it over and over again, and I am still not entirely sure whether or not I am missing something. In lines 244–245, the authors have concluded on the basis of their isotopic records for pore-fluid methane and co-existing CO₂, that methane production must have happened through hydrogenotrophic methanogenesis. Based on the corresponding isotopic profiles, I would agree with that statement, and I would add that the increasing carbon isotope values with depth may well be recording an overall kinetic fractionation effect superimposed on equilibrium isotopic fractionation between the two species across the sediment column. Here is the issue though: please excuse my utter naivety if I have let it slip in again, but my understanding of hydrogenotrophic methanogenesis is that the reactants are CO₂ and hydrogen, and that no organic matter per se is implicated! Furthermore, given that the budget of DIC in the sediments is quite high to drive siderite saturation, and the carbon isotope value of pore-fluid CO₂ it is as low as -15 per mil, then surely another pathway of organic matter respiration must have preceded methanogenesis to deliver such light CO₂ into pore fluid? Again, the overarching conclusion is that this study is crying out for carbon isotope analyses of the contained siderite to be performed, as a

potentially invaluable proxy of what is really going on, and crucially, whether what is going on is consistent to what might also have happened during BIF deposition and diagenesis. So, to conclude: whereas I have no reason to dispute the validity and robustness of the authors observations and data generation, I am not at all convinced by their interpretations, nor by the general analogy of these lakes with the early ferruginous oceans, for that matter. I should also perhaps add here that I am sensing a bit of an "urge" by the authors through this paper, to contrive methanogenesis as the next big thing in BIF-related research. If they care about my own opinion, I see absolutely no obvious parallels that can be drawn between the lakes and the BIF ferruginous ocean of the Neoproterozoic, on the basis of methane formation as a common denominator; this manuscript, at least, does not quite "cut it" in that respect. My verdict will therefore have to be rejection I'm afraid, but my suggestion remains: analyze the siderites for carbon isotopes and re-assess! After all, that is the only direct proxy of carbon cycling the research community can resort to, if they are to constrain the fate of carbon during formation of BIF...

Reviewer #3 (Remarks to the Author):

This paper from Friese and colleagues describes geochemical profiles from sediment cores in ferruginous Lake Towuti. The authors suggest that whilst it would be expected that iron reduction dominates organic carbon mineralization, methanogenesis is in fact the dominant pathway. Their explanation hinges on the fact that although iron is abundant in the sediments, it is not reactive enough for use by iron reducing microorganisms. The majority of organic carbon turnover is therefore driven by methanogenesis. The authors use this data to suggest that methanogenesis would be a significant process in ferruginous sediments, and that the extent of the contribution of methanogens was linked to Fe reactivity.

This study provides novel insights into organic matter turnover in unusually ferruginous sediments and provides some clues as to what processes may have dominated in ancient ocean environments.

The paper will certainly be of interest to those studying marine sediments, and the community of researchers interested in how sedimentary and atmospheric processes may have been linked in Earth's history. It will certainly influence the thinking in the field, particularly in regard to the importance of Fe reduction in ferruginous sediments.

Most of the claims are convincing, although there are some areas where clarification or additional discussion are required:

- 1) There are no errors included in figure 2, 3 or 4, or figure S2, S3 or S4 which represent all of the main geochemical and incubation data. It is unclear whether they were omitted for clarity, whether they are smaller than the symbols or whether there was no replication of these measurements.
- 2) The rationale used to separate iron into different fractions are not always clearly explained. I have listed a number of queries regarding this below.

There are also two potential gaps in the data set in that both Fe mineralogy and microbial processes are always inferred, but never directly observed. To strengthen the interpretations in the manuscript it would be great if the authors could consider the possibility of incorporating the following analyses:

- 1) **Fe mineralogy:** Since the Fe mineralogy is so central to the story it would significantly improve the manuscript to have some direct measurement. The sequential extractions are fine (and a lot of effort which I appreciate!), but they do not give any real specificity

on what the mineral content actually is. Given the high iron content, these seem like prime candidates for Moessbauer spectroscopy.

- 2) **Microbial community:** Some discussion of what is known about the microbial community in the core would be extremely beneficial since all interpretations rely on microbial processes. The authors write that the core was collected specifically for Geomicrobiological analyses which suggests samples suitable for this kind of analysis would have been preserved.

I appreciate these are not minor tasks. If it is beyond the capability of the authors to explore this, perhaps there is data in the literature from other studies in Lake Towuti which would allow for some comment on this.

Overall, this is an interesting study where the claims are generally supported by the results. I would be very happy to see this ultimately published.

Specific comments

Line 134: Highlight in the text that siderite formation in Table 3 is modelled and not measured

Line 139: Highlight that this is total *extractable* iron concentration

Figure 2: There is no error associated with any of the data in figure 2. The iron extractions especially can be quite variable as small differences in extraction time for each step can influence the amount of iron extracted. There is no indication in the methods how much replication was included for these extractions.

Line 149: Clarify whether the “total iron pool” referred to here is the sum of the Fe concentration in all fractions or whether the 30% figure stated the percentage of the 0.5M HCl extractable iron.

Line 161: Dithionite is a poor extractant for nontronite (< 5% reported by Poulton & Canfield in their original iron extraction protocol). The text should be modified to highlight this. Nontronite will show up in the fraction which was boiled in strong HCl.

Line 162: Suggest also stating “880 $\mu\text{mol cm}^{-3}$ ” as a percentage of total iron as done for the other fractions.

Line 162: How was the value stated for dithionite-extractable iron quantified? From the text it reads like the Fe(III) in the extraction was quantified, but that would not make sense for an extraction based on a reductive dissolution i.e. all Fe should be Fe(II) in the extract regardless of its original redox state.

Line 164: The Fe_{oxa} extraction which will extract magnetite is not necessarily only contributing to the reactive Fe(III) as magnetite is mixed valent. It is not clear from the description in the methods whether both Fe(II) and Fe(III) were quantified in all extracts or only in the selected ones mentioned.

Line 166: How significant is this decrease in dithionite-extractable Fe in the upper one cm? Without any error this is hard to tell.

Line 172: It reads here like the authors would expect a decrease in both aqueous Fe(II) and pH when Fe reduction was occurring. Can they expand on why this is the case. The pH profiles should also be referenced here (they are not in figure 4).

Line 172: Can the authors expand on the extent to which it would really be expected that pH is significantly altered by Fe reduction in such a well-buffered system?

Figure 4: How do the authors explain the coinciding increase of Fe^{2+} and acetate, and decrease in lactate at around 4-6m. Considering that iron reducers convert lactate to acetate and Fe(III) to Fe(II), is this not at least circumstantial evidence for iron reduction here?

Line 192-193: In general I think the evidence is quite strong that Fe reduction rates are low, but I am not entirely convinced (yet) that the Fe reduction is entirely limited to the surface sediment due to the observations in my comment on Figure 4. It also seems to me that the claim that the Fe(III) minerals are stable over tens of thousands of years is a bit of a stretch based on the available data which simply compares how reactive the minerals there are, but cannot show whether these are the same type of mineral.

Line 202: The authors highlight here that whilst the geochemical data and modelling do not point to sulfate reduction, the radiotracer experiments show there is sulfate reduction. When their claim that

iron reduction does not occur at depth is exclusively based on geochemical data, can they rule out a similar process for iron?

Line 278: The authors suggest crystallinity of the Fe minerals leads to inhibited Fe reduction and the out-competition of Fe-reducers by methanogens, but can it be ruled out that the availability of potential electron donors for iron reduction does not also play a role? Fe-reducers and methanogens have some substrates in common (e.g. acetate, H₂) but their flexibility towards other substrates is different. The section would benefit from some discussion of other potential drivers of the lack of Fe reduction, in addition to the crystallinity.

Line 839-849: The authors should supply additional information here on how the Fe(II) and Fe(III) were calculated.

REVIEWER COMMENTS

Reviewer #1 (Remarks to the Author):

This paper documents the biogeochemistry of Fe-rich, S-poor, permanently anoxic sediments of Lake Towuti in Indonesia, with the goal of evaluating the importance of microbial Fe(III) oxide reduction versus methanogenesis in sediment carbon metabolism. Previous work in analogous, nearby Lake Matano concluded based on water column methane profiles and mass balance calculations that methanogenesis is likely to be the dominant anaerobic carbon oxidation pathway (Crowe et al., *Geobiology*, 2011). However, to date no studies have reported a detailed analysis of Fe geochemistry in the sediments of any of the ferruginous lakes in Indonesia, i.e. measurements that could constrain more directly the potential contribution of Fe(III) reduction to sediment carbon metabolism. Thus, the paper is timely and advances our understanding of the biogeochemistry of these unusual modern ecosystems.

More broadly, the ferruginous Indonesian lakes provide insight into biogeochemical processes in Precambrian marine systems, i.e. prior to and after the Great Oxidation Event when Earth's oceans were likely rich in Fe(II) (and low in sulfate), and when oxidation of Fe(II) in surface waters likely led to the delivery of large quantities of Fe(III) oxides to surface sediments, where they could have served as electron acceptors for sediment carbon oxidation. In fact, the main underlying goal of this paper is to use the results from Lake Towuti to place constraints on the extent to which Fe(III) oxide reduction may or may not have significantly influenced the production and release of methane from Precambrian marine sediments. An enormous body of literature has been devoted to understanding the signatures of microbial Fe(III) oxide reduction in Precambrian marine sediments (e.g. ones that led to Banded Iron Formations), and this paper seeks to expand to our understanding of the potential quantitative role of this pathway, using the Fe-rich sediments of Lake Towuti as a modern analog. In fact, there is a real paucity of such modern analogs, owing to the high abundance of sulfate in modern oceans which ultimately causes the sulfur cycle (i.e. sulfate reduction) to impinge on sediment Fe cycling even in situations where Fe is relatively abundant.

The authors present a comprehensive set of biogeochemical measurements across the upper 10 m of sediment, including: dissolved (porewater) Fe(II); sequential selective extractions to speciate solid-phase Fe; sediment sulfate distribution and radiotracer sulfate reduction rate measurements; dissolved methane concentrations; $\delta^{13}\text{C}$ values for dissolved methane and dissolved inorganic carbon; and concentrations of short-chain organic acids that serve as electron donors for anaerobic microbial respiration. To summarize, both mass balance calculations and sediment geochemical modeling suggest that methanogenesis is in fact that dominant pathway for organic carbon metabolism in these permanently anoxic sediments. Although "reactive" Fe(III) oxide phases (i.e. those released via extraction with dilute HCl, ammonium oxalate, and/or citrate/sodium dithionite) are present in relatively high abundance (hundreds of $\mu\text{mol cm}^{-3}$), they show no systematic decline with depth below the top 1 cm of sediment. In addition, sulfate concentrations are near background throughout the sediment column, such that the potential contribution of sulfate reduction to carbon metabolism is minimal. Hence, as inferred previously for Lake Matano, this study demonstrates explicitly and convincingly that the vast majority (ca. 90%) of sediment carbon metabolism is mediated by methanogenesis in the ferruginous lakes of Indonesia.

The results from the ferruginous Indonesian lakes have important implications for sediment carbon metabolism in what are assumed to be ferruginous Precambrian marine systems. However, a key issue regarding the use of the Indonesian lakes as direct analogs to Precambrian marine sediments relates to nature and origin of the presumably "reactive" – but apparently non-microbially-reducible – Fe(III) oxide phases that persist in these Fe-rich sediments. In the introduction to the paper the

authors state that “Intensive weathering of ophiolitic bedrock from the catchment supplies the lake with a strong influx of iron(oxy)hydroxides”. The problem here is that this source of Fe input is fundamentally different from the source of Fe(III) oxide that was likely delivered to Precambrian marine surface sediments. In the case of the latter, it was almost certainly oxidation of Fe(II) in the water column (e.g. via either anoxygenic photosynthesis or reaction with oxygen produced by oxygenic photosynthesis), rather than inputs from continental weathering, that was the dominant source of Fe(III) oxide input. Given the relatively high concentrations of dissolved silica in Precambrian oceans, it seems likely that the Fe(III) oxide phases delivered to the sediments were actually oxide-silica coprecipitates, which experimental studies have shown to be subject to near-complete reduction by dissimilatory Fe(III)-reducing bacteria (Percak-Dennett et al., *Geobiology*, 2013; Fortney et al., *Geobiology*, 2016). It seems possible that the large quantities of non-reducible, crystalline phases such as goethite and hematite represent mainly inputs from the surrounding catchment rather than oxidation of Fe(II) in the water column. In addition, even Fe(III) oxides produced via anoxygenic photosynthesis (evidence for which, notably, was provided by Crowe et al., *PNAS*, 2008 in Lake Matano) or reaction with oxygen in the photic zone could result in relatively rapid transformation to crystalline phases in the absence ligands such as phosphate (commonly present in high abundance in microbiological culturing experiments) or silica that are well known to retard Fe(III) oxide crystallization during Fe(II) oxidation.

The upshot of the above considerations is that the quantitative extension of the results from Lake Towuti and ferruginous Indonesian lakes to Precambrian oceans needs to be qualified with respect to the microbial reducibility of the Fe(III) oxide phases. To achieve this, the authors consider end-member scenarios where the Fe(III) oxides deposited to Precambrian marine sediments are assumed to have either high or low reducibility. The high reducibility scenario is based on results from Chocolate Pots hot spring in Yellowstone National Park, where Fe(III) oxide-silicate coprecipitates (formed where Si/Fe(II)-rich neutral-pH groundwater is discharged to the land surface) are known to be subject to extensive (ca. 80%) microbial reduction. The low reducibility scenario is based directly on the mass balance results from Lake Towuti sediments. In doing this, they are able to constrain the potential contribution of microbial Fe(III) oxide reduction versus methanogenesis to sediment carbon metabolism in Precambrian marine sediments with varying degrees of Fe(III) oxide reducibility. The results suggest that increased oxide reducibility, as might have been expected for Fe(III) oxide-silica coprecipitates formed in the water column, could have led to substantial decreases in the amount of methane released from Precambrian marine sediments. The authors incorporate a discussion of the average oxidation state of Fe oxide mineral phases in Precambrian sediments, which is useful for “tethering” their results to the geological record. The overall conclusion is that variations in Fe(III) oxide reducibility, and hence preserved Fe mineral oxidation state, could have varied across facies, between deposits, and through time, with attending potential impacts on sediment carbon metabolism and methane production.

In my opinion, this is a well-developed body of work that will be of broad interest to biogeoscientists interested in both modern and ancient ferruginous aquatic systems. The conclusion drawn for Lake Towuti (and by extension other analogous ferruginous Indonesian lakes) are well supported by the data, and the authors do a nice job of explaining things in a succinct, interesting manner. They also succeed in connecting the results from this modern environment to analogous Precambrian marine systems, thereby contributing to our evolving understanding of what controlled atmospheric methane concentrations before and after the Great Oxidation Event.

RESPONSE: *We thank reviewer #1 for recognizing the quality, importance, and impact of our manuscript and the succinct but detailed description of our work.*

Reviewer #2 (Remarks to the Author):

It was a privilege to be asked to review the manuscript by Friese et al. entitled: “Iron oxide reactivity controls organic matter mineralization in ferruginous sediments”, submitted for consideration to Nature Communications. First and foremost, my sincerest apologies must be conveyed to everyone concerned with this submission, for the delay in turning in my review. I fell ill recently and that set me back considerably, at a time when I am also overloaded with remote teaching responsibilities. The fact that I am based at a southern hemisphere university and thus find myself in the middle of the academic year, did not help the cause much either.

I will elaborate a bit in my review, even though I never thought I would when I first received the manuscript. This is because I am recommending rejection of it, and my reasoning is based on a couple of truly fundamental shortcomings of this work which I cannot see how they can be adequately rectified through any kind of revision to the current version. In the following lines, I will try and provide an insightful enough justification for my recommendation, with the hope that the authors and the editors will see my concerns and my points clearly enough. It is important to note that I read and assessed this work from my own specific viewpoint of someone who has worked on BIFs all their (scientific) lives. Therefore, my critique will revolve, in most part, around how the results and the arguments that the authors present, speak (or do not) truly and directly to the early Precambrian ferruginous ocean, and to the putative processes at play in that environment.

After all, the discussion of the authors places its largest emphasis on precisely that aspect, given that lake Towuti and nearby lake Matano are variously considered as modern analogues of Precambrian ferruginous oceans and, by extension, of the iron-rich sediments that precipitate/d in them.

RESPONSE: *Whereas we use Lake Towuti as our field site, our study, however, deals with fundamental biogeochemical processes in ferruginous systems, irrespective of a specific location. It is these processes themselves that are the analogues rather than the host environments.*

The first point I wish to raise might come across as a minor one, but is nonetheless strongly indicative of the direction that I am heading towards in this report: the statement in lines 68-70 about Fe-rich shale being the dominant sediment type in the early ferruginous ocean while BIF was an “extreme” end-member of iron deposition, is rather postposterous, if I may say so.

RESPONSE: *The reviewer stated above that they have worked on BIFs all their (scientific) lives and we very much appreciate this perspective. In our view, this particular comment partly reflects a misunderstanding related to terminology. Here our intention was to contrast the deposition of Fe-rich chemical sediments to more siliciclastic-rich sediments. We recognize that such siliciclastic sediments are themselves also components of BIFs that can be interlayered with pure or near pure chemical sediments. For example, half of the classic Hamersley BIFs consist of shaley sediment (the so-called S-bands) and BIFs were only deposited over a fraction of continental shelves even during intervals of intense deposition, which itself was only intermittent. This is evident in the classic Hamersley Basin stratigraphy where the BIF does NOT extend across the entire shelf and is absent in the far east Pilbara. While we acknowledge that Algoma-type BIFs were formed around hydrothermal systems in deeper water (e.g. Bekker et al., 2010, Econ. Geol.), the quantitatively much more important superior type BIFs (of which Hamersley is a prime example) likely formed in shelf/slope like environments (Konhauser et al., 2017, Earth. Sci. Rev). The greater tendency for shallow water deposits to be accreted and thus preserved in the geological record would also lead to bias in preservation. There is a strong bias towards studying BIFs for many reasons and there are many more papers on BIFs than on siliciclastic sediments from these Eons, and we are not convinced that the number of publications*

about a specific rock type should be used as an indication of its natural abundance. We also note that mass balance likely requires that BIF deposition was restricted to small parts of the ocean, whereas siliciclastic and carbonate deposition must have characterized much of the remaining ocean floor. See for example Thompson et al. (2019a) or compare the numbers of Husson & Peters (2018) about global sediment volume through time with the known volumes of BIFs. We altered the text accordingly, and the specific text noted above now reads:

Line 71: Precipitation of Fe from these oceans resulted in the widespread deposition of ferruginous siliciclastic sediments and, in cases, nearly pure chemical sediments like those characteristic of many banded iron formations (BIFs) ⁸.

I have read several times before the solitary citation accompanying that statement (i.e. Poulton & Canfield, 2007) and I did not quite see any support for such a contention;

RESPONSE: *Perhaps we made an error initially, however, we are unaware of the paper Poulton and Canfield (2007), and it does not appear in our reference list. With respect to ferruginous conditions, we cite Poulton & Canfield (2011), as it provides a good overview of the topic. We could have also cited, Walker (1987) who describes BIF deposition under highly productive zones (which we interpret as analogous to modern upwelling systems, e.g. Thompson et al. 2019), or Isley & Abbott (1999) who illustrated that BIF deposition was episodic and linked to intervals of enhanced volcanism, or Reinhard et al. (2009) who discussed deposition of Archean ferruginous shales and episodes of euxinia. We have, nevertheless, altered the statement linked to the citation, as noted above.*

even if that was the case though, one wonders why the authors later in their Discussion, interpret their results exclusively in the context of the “extreme BIFs” and not of the “widespread ferruginous shales” they are referring to in their introductory lines.

RESPONSE: *It was never our intention to discuss our results exclusively in the context of BIF, as our findings would equally apply to ferruginous siliciclastic sediments. We have revised the manuscript to make this clearer. We also tried to avoid the term shales and use siliciclastic sediments now, to keep a broader perspective.*

Moving to the section "Sedimentary iron phases" (lines 138-167), the authors present geochemical results that are essentially the only proxy of the true mineralogy of the studied sediments.

RESPONSE: *Our view here it is the distribution of Fe between pools that have characteristic chemical/biological reactivities is most important to our claims, and in our view the extraction scheme is well suited to this. As the reviewer notes, mineralogy is nevertheless informative and we have thus added new XRD analyses to directly tether the extraction data to mineralogy (Fig. S4, Table. S5). Importantly, this confirms the abundance and stability of goethite throughout the sediments and it's almost quantitative extraction in the dithionite step as targeted.*

Figure S4: *XRD spectra for the upper 12 m of Lake Towuti sediment. The main reflectance peaks for goethite are identified (labeled vertical grey lines “G”), which reveal goethite in every sample.*

Through standard sequential extraction, they have derived information about ferric iron availability in the sediment, that would constitute the “reactive iron pool” for organic carbon re-mineralisation. The result is quite astounding: the “easily” soluble fraction (i.e. that derived through acetate and 0.5M HCl digestions) returned no ferric iron whatsoever, and instead revealed the exclusive dominance of Fe(II), which makes up approximately one-third of the total iron pool! This, to me, means no ferrihydrite preservation whatsoever, unless (for some reason) the primary ferrihydrite did not react in weak HCl but only later during oxalate/dithionite extraction.

RESPONSE: *The reviewer is entirely correct and there is no preservation of ferrihydrite or ferrihydrite-like phases below the uppermost sediment layer. To make this abundantly clear we have rephrased some parts of the text to stress that ferrihydrite or ferrihydrite-like phases are entirely*

exhausted through respiration in the very uppermost sediments (top 1 cm or so) or even in the water column, as shown in previous studies in both lakes Matano and Towuti.

Line 161: *Despite the high abundance of acetate extractable Fe(II) and pore water supersaturation, siderite only accumulates in sporadic carbonate-rich layers (Vuillemin et al., 2019), and mineral carbon is low throughout most of the sediment (Ordoñez et al., 2019). Much of this acetate extractable Fe(II) and by extension mineral carbon and siderite likely forms in response to Fe(II) reduction in the water column (Bauer et al., 2020a), as previously shown.*

Line 195: *These observations thus reveal that a large fraction of the microbial Fe reduction in Lake Towuti is restricted to the water column (Bauer et al., 2020a) and uppermost sediment layer.*

Line 377: *Mineralogical and isotopic observations, in fact, are increasingly pointing towards primary deposition of Fe(II) minerals with a comparatively minor role for diagenetic Fe(III) reduction (Canfield et al., 2018; Siahi et al., 2020), which is in line with the primary locus of Fe(II) reduction and mineral formation in Lake Towuti's water column.*

This, however, is not at all clear, as there is no information presented by the authors that would constrain which phases exactly yielded ferric iron upon those two digestion steps.

RESPONSE: *Again, our view is that mineralogy (we presume “phases” refers to mineral phases) is perhaps less important than reactivity, which we assess via sequential extraction and we therefore prefer the term “fractions”. However, we thank the reviewer for pointing out the need for clarity on this and added Table S5 to the Supplementary Information with detailed results of the sequential extraction and speciation data.*

In fact, in the absence of any added information on the Fe(II)/Fe(III) ratio (I presume that the authors did titrate their sample solutions in order to be able to determine the oxidation state of Fe in the earlier extractions), one could conceivably argue that the ferric iron may have been dominated by magnetite dissolution, which means that Fe(II) might still be “lurking” here too.

RESPONSE: *We agree with the reviewer that the oxidation state of Fe in rocks can be determined through titrations, however, we used well-established method, to determine or infer iron speciation and the Fe(II)/Fe(III) ratio of each fraction. As dithionite selectively dissolves Fe(III) minerals by reduction of Fe(III) to Fe(II), the resulting Fe in solution is thus 100% Fe(II). We assume that the vast majority of the dithionite fraction is made up by goethite and hematite, which has been abundantly verified (Poulton and Canfield, 2005) and therefore consider all Fe liberated by the dithionite extraction step to be Fe(III). This is further supported by our new qXRD results (Table. S5). Magnetite, of course, contains both Fe(II) and Fe(III), but is not appreciably dissolved in dithionite (Poulton and Canfield 2005).*

Line 430: *Sample masses of 200 mg of sediment were weighed into 15 mL centrifuge tubes, and subjected to the Fe-speciation sequential extraction scheme based on the protocol of Poulton and Canfield (Poulton and Canfield, 2005):*

- *Leaching the sample with 0.5 N HCl for 1 h extracted the highly reactive Fe in hydrous ferric oxides like ferrihydrite or lepidocrocite (Fe_{HCl}). Iron speciation was determined spectrophotometrically (Thamdrup et al., 1994; Viollier et al., 2000). However, easily reducible ferric iron within the Fe_{aca} fraction was always below the limit of detection in all samples, so that this fraction entirely consists of ferrous iron.*
- *Fe(II) in carbonate and poorly crystalline phases was extracted by leaching the sediment with a sodium acetate solution adjusted with acetic acid to pH 4.5 with acetic acid for 48 h (Fe_{aca}).*
- *Reactive Fe(III) in goethite and hematite was extracted by treating the sediment sample with a sodium dithionite solution (50 g L^{-1}) buffered to pH 4.8 with 0.35 M acetic acid/0.2 M sodium citrate for 2 h (Fe_{dith}). Due to the reductive dissolution in this step, the resulting extract only contains Fe(II). However, as the main minerals dissolved in this step only contain Fe(III) we consider this pool to consist only of Fe(III). We further verified the selectivity of the extraction by performing a dithionite leaching experiment on two samples from the upper 30 cm of the Towuti sediment. We first quantified via qXRD the amount of goethite in these samples, and then subsequently leached them with dithionite. Our results (Table S6) demonstrate that within error of the qXRD and*

Fe-speciation measurements, dithionite quantitatively extracts goethite in the Lake Towuti sediments. Furthermore, XRD data demonstrates the presence of goethite throughout the core (Fig. S4), and thus we conclude that Fe contained in our dithionite extractions was indeed goethite.

- *Leaching the sediment sample with 0.2 M ammonium oxalate/0.17 M oxalic acid solution (pH 3.2) for 6 h extracted the reactive Fe present in magnetite (Fe_{oxa}). Given the mixed valence of Fe in magnetite, 2/3 of the extracted Fe was counted as Fe(III) and 1/3 as Fe(II).*
- *Finally, samples were subjected to a near boiling 6 N HCl extraction for 24 h to extract the remaining unreactive Fe in silicates and Fe-bearing clays like nontronite (Fe_{sil}). Fe(II)/Fe(III) ratios were not determined.*

After all, magnetite dissolves (at least partially) in both oxalate and dithionite, and some of the authors of the current study have previously reported magnetite associated with the omnipresent siderite (Vuillemin et al, 2019).

RESPONSE: Only very limited amounts ($\pm 5\%$) of magnetite dissolve in the dithionite extraction step (Poulton and Canfield 2005), the rest is almost quantitatively extracted in the subsequent oxalate step. We therefore do not consider magnetite to be a major contributor of iron of either redox state to the dithionite fraction, especially given the lower (typically ~ 1 wt. % Fe) abundance of magnetite compared to goethite (typically ~ 2 wt. % Fe).

We also respectfully disagree with the reviewer about the omnipresence of siderite attributed to Vuillemin et al. (2019). Siderite is not omnipresent, as siderite is mainly found in discrete layers (We identified many discrete siderite-rich layers in the sediment during our initial core description). Regarding the association of siderite and magnetite, Vuillemin et al. (2019) stated that the TEM analysis “....also identified minor quantities of carbonate GR and magnetite within siderite crystals at 6.2 m sediment depth“. In our present manuscript, we focus on the quantitative contribution of iron-phases of different reactivity to carbon mineralization, whereas Vuillemin et al. (2019) was a qualitative study on siderite formation. Total inorganic carbon concentrations (Russell et al. 2020), clearly show that siderite represents a minor fraction of the total Fe pool, and regardless, does not change the observation that Fe(III) in the form of goethite persists throughout the sediment.

Ref:

Russell, J. M. et al. The late quaternary tectonic, biogeochemical, and environmental evolution of ferruginous Lake Towuti, Indonesia. *Palaeogeogr. Palaeoclimatol. Palaeoecol.* **556**, 109905 (2020).

The truth, therefore, is this: if one were to dismiss the unreactive fraction of the sediments as representing recalcitrant, iron-bearing silicate detritus from the hinterland, then the actual budget of iron yielded through the speciation analyses is dominated by more than 50% Fe(II)!

RESPONSE: The reviewer is correct, but that still leaves a large fraction of the total Fe pool to canonically reactive Fe(III) phases like goethite, which is exactly the point of our manuscript. These canonically reactive Fe(III) phases are not actually reactive towards Fe-reduction in Lake Towuti. Instead, they are recalcitrant (regardless of origin) and OM degradation is channeled through

methanogenesis. Basically, all 0.5 N HCl extractable Fe is reduced in the water column and upper 1 cm of sediment and the rest of the reactive Fe(III) pool remains untouched downcore as shown in the Bauer et al. 2020 EPSL paper, which we now cite here:

Line 161: *Despite the high abundance of acetate extractable Fe(II) and pore water supersaturation, siderite only accumulates in sporadic carbonate-rich layers (Vuillemin et al., 2019), and mineral carbon is low throughout most of the sediment (Ordoñez et al., 2019). Much of this acetate extractable Fe(II) and by extension mineral carbon and siderite likely forms in response to Fe(II) reduction in the water column (Bauer et al., 2020a), as previously shown.*

The very bulk of that iron ought to have been hosted in siderite, given both the current results and the recent study of Vuillemin et al. (2019) that deals with exactly that phase in the sediments and its possible origins.

RESPONSE: *Some of it is siderite, but for our purposes it doesn't matter. Based on TIC concentrations measured via coulometry (Russell et al., 2020), siderite is only found in discrete intervals and represents a minor fraction of the bulk iron, which is in agreement with the findings of Vuillemin et al. (2019), as previously discussed above.*

In view of the above, the question I am readily asking is this: how did that siderite exactly form? Unfortunately, the same paper (Vuillemin et al., 2019) does not provide a clear-cut pathway of siderite formation, and instead proposes a couple of possibilities, such as diagenetic replacement of carbonate green rust. (It should be noted here that siderite formation via calcic carbonate replacement during diagenesis has also been reported elsewhere: see Lin et al., Geobiology, 2019).

RESPONSE: *Siderite is supersaturated throughout the sediment pile and can thus form anywhere really, but given that Fe reduction is restricted to the water column and very uppermost sediments, it likely forms there in response to Fe(II) production and corresponding accumulation of inorganic carbon and the carbonate ion. While siderite is supersaturated throughout the core, calcite is undersaturated, as shown in (Table 1) in Vuillemin et al. (2019), and thus siderite formation through carbonate replacement seems less likely in Lake Towuti sediments than in Precambrian seawater .*

If that is the case though, then one needs to really delve into the specifics of the ultimate source of Fe(II) in this stratified lake system: if the authors want to argue that there is methane generation in the sediment pile but no anaerobic oxidation thereof by ferric iron or sulphate, then something must be reducing iron, and that reduction must be efficient enough to lead to saturation in siderite!

RESPONSE: *See previous comments, but indeed it is organic matter that reduces Fe in the water column and near the sediment water interface. In our manuscript we describe in detail that there is abundant organic matter with capacity to reduce all available electron acceptors.*

Line 124: *Lake Towuti's sediment is relatively rich in organic carbon (TOC 0.4-4 wt%) with elemental compositions implying that it is reactive (molar C:N ratio 11-25), readily fermentable (Fig. 2), and consequently reactive towards microbial respiration.*

We also expanded the text to include a statement about pore water being supersaturated with respect to siderite throughout the core.

Line 159: *Concomitantly high DIC and Fe(II) concentrations translate to pore waters that are supersaturated with respect to siderite (FeCO₃) throughout the sediment (Table S3) (Vuillemin et al., 2019). Despite the high abundance of acetate extractable Fe(II) and pore water supersaturation, siderite only accumulates in sporadic carbonate-rich layers (Vuillemin et al., 2019), and mineral carbon is low throughout most of the sediment (Ordoñez et al., 2019). Much of this acetate*

extractable Fe(II) and by extension mineral carbon and siderite likely forms in response to Fe(II) reduction in the water column (Bauer et al., 2020a), as previously shown.

That saturation would have to be achieved either in the water column to produce primary siderite precipitates (a possibility also discussed by Vuillemin et al., 2019), or in the sediment, to justify replacement of other primary carbonate material by siderite.

RESPONSE: *Again, the Vuillemin et al. 2019 paper is largely qualitative in nature. Siderite formation and recrystallization has little quantitative effect on sediment Fe mass balance (see TIC profile) and is thus not as relevant to this discussion.*

These are all questions (and there are plenty more) that I cannot find any reasonable answers for in the current manuscript. Clearly, establishment of stratification and ferruginous conditions in a lake requires processes of iron redox cycling at play: if that hardly happens in the sediment column, it has to dominate in the water column itself.

RESPONSE: *This is precisely the point. Yes, iron redox cycling is active in the water column (and very uppermost sediment), but hardly in the sediment. This is why an appreciable fraction of ferric minerals in the sediment are preserved over geologic timescales.*

So, is it possible that bottom waters in lake Towuti are generating ferrous iron by reduction of primary ferrihydrite? If they do, what is the electron donor? Is it organic matter? In other words, is iron reduction as effective here as to happen already in the water column, in marked contrast to the arguments of the authors?

RESPONSE: *This is indeed the case (see Bauer et al. 2020, EPSL), and we have tried to emphasize this in the revised MS. All Fe reduction takes places in the water column and very upper reaches of the sediment, whereas organic matter degradation in the sediment is largely channeled through methanogenesis. We have expanded the start of the discussion and added the Bauer et al. (2020) reference.*

Line 195: *These observations thus reveal that a large fraction of the microbial Fe reduction in Lake Towuti is restricted to the water column (Bauer et al., 2020a) and uppermost sediment layer. At the same time, this also reveals that the Fe(III) phases found throughout the sediment studied are stable for over tens of thousands of years.*

Line 377: *Mineralogical and isotopic observations, in fact, are increasingly pointing towards primary deposition of Fe(II) minerals with a comparatively minor role for diagenetic Fe(III) reduction (Canfield et al., 2018; Siahi et al., 2020), which is in line with the primary locus of Fe(II) reduction and mineral formation in Lake Towuti's water column.*

I have to digress a bit here in order to add that current classic models for BIF diagenesis as put forward by the Wisconsin and Tubingen groups over the years, invoke dissimilatory iron reduction as THE key pathway of organic matter cycling. The same models also predict that DIR would have operated exclusively below the sediment-water interface, and thus make no provision for it in the ambient water-column. Therefore, the BIF diagenetic models seem to be diametrically incompatible with what is interpreted by the authors to be happening in Lake Towuti, raising serious doubts as to how appropriate an analogue it is for ferruginous oceans in deep time.

RESPONSE: *This is the point of the Bauer et al., (2020) and to some extent portions of the Thompson et al., (2019) papers. The point here is that abundant ferric Fe is preserved. Clearly, the current models for BIF formation need to be revised, based on our new information, to consider that an appreciable fraction of the Fe(II) preserved in BIF must be sourced directly in the water column. This is consistent with C-isotopes in BIF siderite, which require appreciable precipitation in the water column rather*

than the sediment (see discussion in Thompson et al. 2019, for example). We thank the reviewer for pointing this out as it motivated us to better place our results in the context of emerging narratives around the role of primary deposition of Fe(II) and mixed-valence minerals in the formation of BIF, which is highlighted in the revised MS.

Line 377: Mineralogical and isotopic observations, in fact, are increasingly pointing towards primary deposition of Fe(II) minerals with a comparatively minor role for diagenetic Fe(III) reduction (Canfield et al., 2018; Siah et al., 2020), which is in line with the primary locus of Fe(II) reduction and mineral formation in Lake Towuti's water column.

An alternative electron donor of course for water-column Fe cycling may have been the methane itself that the authors report: should one be led to assume then, that methane is not oxidized in the sediment but it may endure near-quantitative oxidation in the water column, thus leading to conditions favoring siderite saturation?

RESPONSE: *The actual importance of Fe-driven methane oxidation in natural systems remains unresolved, but must be small. We would therefore like to clarify that Fe is an unlikely oxidant for methane in any natural environment. There is a wealth of anecdotal and circumstantial evidence for Fe(III) driven methane oxidation, but so far there is no robust evidence that this process has any quantitative importance in natural settings. The most frequently cited studies in this context are either laboratory studies (e.g. Ettwig et al., 2016, PNAS, Sivan et al., 2014, PNAS; Bar-Or, 2017, EST) or show that, when compared to sulfate-driven AOM, this process does not have much of a quantitative importance in natural sediments (e.g. Beal et al., 2009, Science). Note that the strong laboratory evidence for Fe(III) driven methane oxidation (Ettwig et al., 2016 PNAS) requires high (mM) concentrations of Fe(III)-citrate not relevant to most natural environments and certainly not to Lake Towuti.*

In sum, we conclude that there is no detectable methane-driven Fe(III) reduction downcore in Lake Towuti, despite juxtaposition of high concentrations of Fe(III) (as dithionite extractable phases like goethite or hematite) and mM methane concentrations. Our findings thus argue strongly against Fe(III)-driven methane oxidation in-situ in real-world ferruginous environments. We have added sentences discussing this to the Main Text.

Line 48: *Coexistence of ferric iron with millimolar concentrations of methane further demonstrates lack of iron- dependent methane oxidation.*

Line 300: *With nitrate and nitrite below our limit of detection (4 μ M) and very little sulfate, Fe-dependent AOM remains the only pathway with potential to quantitatively consume methane in Lake Towuti's sediment. While it is true that tightly coupled methane production and oxidation could be masked in pore water profiles (Beulig et al., 2019), the process would cause a decline in oxidant concentration with increasing depth, in this case Fe(III), which we do not observe. Pore water profiles also show no evidence for net methane consumption and Fe-reduction rates are small, so we thus conclude that AOM is generally negligible in Lake Towuti's sediments, as the known electron acceptors are either not available (NO_3 , NO_2 , SO_4) or not utilized (Fe(III)).*

The implications of such an interpretation are of course truly profound in the context of BIF, for more than one reasons: for example, coupled methane and iron cycling in the water column would result in the ongoing introduction of isotopically very light carbon into the DIC budget.

RESPONSE: *The implications are indeed profound and this why we aim to publish in a high impact journal like Nature Communications. However, they are profound because they imply a likely lack of methane reoxidation in the absence of appreciable sulfate and oxygen in the Archean. Thus, any methane produced, and as we argue this production would represent a large fraction of the organic matter deposited, would end up in the atmosphere due to lack of methane oxidation.*

We feel that the outsized effects on isotope budgets anticipated by the reviewer for hypothetical Fe-driven anaerobic methane oxidation, are unlikely. This is because the isotope effects would strongly depend on carbon mass balance. In particular, the isotope effects would depend on how carbon cycling operates and reservoir sizes are critical here. For example, a large DIC pool (e.g. via high CO₂ fluxes from volcanism in conjunction with limited photosynthesis) would mute the isotope effects of methane oxidation. There are certainly some important outstanding questions related to Archean isotope budgets and the C-isotope compositions of siderite—like where is the heavy carbon that balances the light carbon in siderites? Further discussion here, however, is well beyond the scope of our paper and this review particularly given that we conclude that methane oxidation coupled to Fe reduction is negligible.

This would definitely be inconsistent with the contention of a Neoproterozoic ocean containing high DIC abundances at near-0 per mil as proposed previously (e.g. see Fischer et al., Precambrian Research, 2009).

RESPONSE: *High DIC (i.e. a large DIC reservoir) would tend to mute the isotope effects of an active methane cycle as described above.*

The other issue, of course, is that if methane found itself into the contemporaneous atmosphere to act as potent greenhouse gas against a dim early sun (Thompson et al., Science Advances, 2019), one key reductant for iron redox cycling – and hence iron carbonate saturation – would be removed from the equation, leaving only organic matter available to “do the trick”.

RESPONSE: *Correct, as some of us argued in the Thompson et al. (2019) paper. We do not see any contradiction to our manuscript in this statement, and in fact, this is precisely why we conclude strong fluxes of the methane to the Archean atmosphere.*

I do not think there is any need for me to continue my thought process as conveyed in the foregoing paragraphs, simply because the possibilities of alternative interpretation are almost endless and the current manuscript fails to provide any meaningful constraints whatsoever towards resolving the various overlapping issues.

RESPONSE: *Respectfully, we disagree here. There are only a few scenarios that work and these are the scenarios advanced in our MS. Simply, sedimentary Fe reduction by organic matter (or methane) is ineffective and this leads to methane production from available organic matter. Lack of iron-driven methane oxidation means methane escapes the sediment. Without any other oxidant (oxygen, nitrate or sulfate) in pore water or overlying waters, this methane would end up in the atmosphere. Such a scenario was likely in the Archean, suggesting strong biospheric methane fluxes to the Archean atmosphere.*

The bottom line is this: on the one hand, the authors posit that diagenetic iron reduction is ineffective to the point where organic matter ends up turning into methane, which then oozes out of the sediment without suffering any measurable oxidation,

RESPONSE: *This is exactly what we propose, and thus this is a succinct description of our key findings.*

and yet the sediment is full of siderite!

RESPONSE: *There is in fact very little siderite and what is there is limited to discrete layers that likely formed in the water column or the very upper reaches (1cm) of the sediment. The sediment instead is full of unreacted Fe(III) phases. This figure clearly shows that while siderite reaches appreciable concentrations in some narrow and discrete deep layers, TIC concentration is in the zero to single percent range through most of the sediment reported in the present study.*

There is a good deal that does not agree with me on this, as I have already argued in the preceding passages of text; another example would be the statement in line 343 that the average iron oxidation in BIF is 2.6, something that not only do I vehemently disagree with,

RESPONSE: *Thompson et al. (2019) calculated the average oxidation state for BIFs (2.6) based on an extensive literature review. The entire dataset can be found in the supplement of that paper. With some exceptions at either end of the spectrum, the bulk of the data is actually quite close to the average.. However, we do acknowledge that very nature of BIFs (i.e., that they consist of alternating bands with different mineralogical compositions) makes calculating an average oxidation state for Fe somewhat misleading. Of course, if one were to examine a siderite rich band, the oxidation state of Fe would be different from bands containing almost exclusively hematite or goethite. We have added a sentence to the manuscript to address this.*

Line 356: *These analogies can be tethered to the geological record by considering the redox state of Fe preserved in BIFs, which on average is 2.6 (Thompson et al., 2019b) (or 2.4 based on other sources, with possible variation from 2.0 for a pure siderite end member, to 3.0 for a hematite end member), which implies that at least 60% of the Fe(III) is commonly preserved and buried.*

but the abundant siderite in the Towuti sediments does too!

RESPONSE: *Please see above.*

Here is, in fact, where my biggest disappointment lies with this manuscript: if there is one proxy I could identify that has huge potential to provide at least some constraints, that is the carbon isotope composition of the siderite itself! To this day, I am completely baffled as to why the authors never did such analyses, or if they did, why they did not include the results here (although I can think of some reasons for the latter, but I would never insinuate that the authors would ever purposely withhold such important data).

RESPONSE: *Siderite is only a minor component of the Fe pool, which likely forms, at least in part, in the water column and very upper reaches of the sediment. That is why it was not included in the original manuscript. We do have siderite carbon isotope data, but it doesn't directly inform on the relevant processes (i.e. sedimentary Fe reduction, methanogenesis), and instead, its heavy composition points to water column formation. The relatively heavy values are also qualitatively consistent with*

an absence of AOM. The variability in the siderite carbon isotopes is harder to explain, especially against the backdrop of limited variability in the isotopic composition of CO₂ in the porewaters, but is likely tied to water column dynamics and thus also points to a primary water column source for siderite. Again, however, this is beyond the scope of the present manuscript.

The carbon isotope composition of BIF siderites and other iron carbonate minerals (i.e. ankerite) is widely documented in literature and is well in the negatives, so it is almost a no-brainer that if one wants to constrain redox cycling of iron and carbon in a modern analogue to a BIF, the very first thing to analyze should be the carbon isotope composition of the contained iron carbonates.

RESPONSE: *Of course carbon isotopes can indeed be informative, particularly in studies where processes like Fe reduction and methanogenesis themselves cannot be directly observed, as is the case for reconstructions based on geologic records, which instead must be based on proxies for the processes. In modern environments, however, there are more direct means of assessing biogeochemical cycling, like through porewater analyses, process rate measurements, and mass balance modeling, as we have done here. While carbon isotopes in siderite are generally informative, they are not useful here because siderite does not play a quantitatively important role in sedimentary carbon mineralization reactions. The lack of siderite production downcore and its relatively minor contribution to the Fe pool combined with the persistence of Fe(III) downcore tell the story, not the isotopes.*

One must also not forget here that, unlike the Towuti sediments which contain percentage level organic carbon, classic BIF is practically organic matter-free globally, and that is precisely the feature exploited in the recent papers of Thompson et al (2019) and Dodd et al (2019), as some of the current authors would be able to attest to. It seems to me that the more one dives into the secrets of the Towuti lake, the less support they will find for its true analogy to BIF depositional systems.

RESPONSE: *The reviewer is correct and there are indeed differences that are important but also illustrative. OM concentrations are higher in Lake Towuti sediments than in BIFs, but they are perfectly analogous to many Archean siliciclastic sediments (see for example compilation in Box 1 from the Lyons et al. (2013), Nature review article). Again, we emphasize that it's not the Towuti sediments that are the analogue, but rather the processes operating therein, and again, the analogy is not drawn to BIF exclusively, but rather to processes operating in Archean (or even Proterozoic) oceans and underlying sediments, siliciclastic and chemical sediments inclusive. Note too that the Thompson et al., (2019) paper (and references therein), call for appreciable siderite formation in the water column rather than exclusively through sediment diagenesis, and it seems that this is an emerging narrative, more generally.*

I will conclude my report through a further reference to the methane-forming process the authors present in their paper. I have read it over and over again, and I am still not entirely sure whether or not I am missing something. In lines 244-245, the authors have concluded on the basis of their isotopic

records for pore-fluid methane and co-existing CO₂, that methane production must have happened through hydrogenotrophic methanogenesis. Based on the corresponding isotopic profiles, I would agree with that statement, and I would add that the increasing carbon isotope values with depth may well be recording an overall kinetic fractionation effect superimposed on equilibrium isotopic fractionation between the two species across the sediment column. Here is the issue though: please excuse my utter naivety if I have let it slip in again, but my understanding of hydrogenotrophic methanogenesis is that the reactants are CO₂ and hydrogen, and that no organic matter per se is implicated!

RESPONSE: This is a great question. *The hydrogen here is ultimately produced through the fermentation of organic matter. The individual reactions can be written in a simplified form:*

For a net reaction:

In reality the process is complicated by the fact that organic matter is more complex and variable than CH₂O and that fermentation produces a number of intermediates like alcohols (fortunately!) and volatile fatty acids, the production and accumulation of which we take as an indication that fermentation is active in Lake Towuti's sediments and that the organic matter there is also reactive.

Furthermore, given that the budget of DIC in the sediments is quite high to drive siderite saturation, and the carbon isotope value of pore-fluid CO₂ it is as low as -15 per mil, then surely another pathway of organic matter respiration must have preceded methanogenesis to deliver such light CO₂ into pore fluid?

RESPONSE:

The reviewer is correct here, and while we don't have sufficient data for a robust mass balance, a back of the envelope calculation implies ~80-85% hydrogenotrophic methanogenesis and ~20-15% respiration, which supports our conclusion of a predominant role for methanogenesis. This calculation assumes reactive organic matter ~-30 ‰, and minimal fractionation on respiration. Note that porewater CO₂ is about -12 to -15 ‰, and correction to DIC, which is the relevant pool for mass balance, gives δ¹³C of DIC in the range of -4 to -5 ‰. With these constraints, the CO₂/DIC produced in response to methanogenesis would be around ~1 ‰. This mass balance is further complicated by open system behavior and variability in δ¹³C of organic matter, but it broadly supports our conclusions.

Again, the overarching conclusion is that this study is crying out for carbon isotope analyses of the contained siderite to be performed, as a potentially invaluable proxy of what is really going on, and crucially, whether what is going on is consistent to what might also have happened during BIF deposition and diagenesis.

RESPONSE: *We respectfully disagree. As explained above, siderite is largely tangential to the story here, and our conclusions are abundantly supported through elemental (and electron) mass balance without need for analyses of siderite.*

So, to conclude: whereas I have no reason to dispute the validity and robustness of the authors observations and data generation, I am not at all convinced by their interpretations, nor by the general analogy of these lakes with the early ferruginous oceans, for that matter. I should also perhaps add here that I am sensing a bit of an “urge” by the authors through this paper, to contrive methanogenesis as the next big thing in BIF-related research. If they care about my own opinion, I see absolutely no obvious parallels that can be drawn between the lakes and the BIF ferruginous ocean of the Neoarchaeon, on the basis of methane formation as a common denominator; this manuscript, at least, does not quite "cut it" in that respect. My verdict will therefore have to be rejection I'm afraid, but my suggestion remains: analyze the siderites for carbon isotopes and re-assess! After all, that is the only direct proxy of carbon cycling the research community can resort to, if they are to constrain the fate of carbon during formation of BIF...

RESPONSE: *There is certainly an emphasis on methanogenesis in our manuscript, its likely importance in ferruginous environments, and thus also in the Archaean oceans. While there is also a connection to BIFs in general (as important deposits from ferruginous oceans), as shown in the Thompson et al., (2019) study, we wouldn't argue that methanogenesis is the next big thing in BIF-related-research. We might argue, however, that water column processes are emerging as key components of the BIF discussion in the geological literature, and our data from Lake Towuti would certainly support this. We also argue that the implications of our work are rather broader than BIFs and instead speak to global biogeochemical cycles and climate regulation across large spans of Earth's history. Again, while isotopic measurement in siderites are interesting in their own right, they are a comparably indirect means of interrogating biogeochemical cycling in modern environments, which instead can be studied through a broad suite of analyses (e.g. pore water chemistry, gas geochemistry, direct turnover rate measurements, etc.) that more directly inform on process. It is true that such analyses are not applicable to studies of rocks from the Archaean Eon, which instead must be addressed through proxy data like C-isotopes in siderites.*

Reviewer #3 (Remarks to the Author):

This paper from Friese and colleagues describes geochemical profiles from sediment cores in ferruginous Lake Towuti. The authors suggest that whilst it would be expected that iron reduction dominates organic carbon mineralization, methanogenesis is in fact the dominant pathway. Their explanation hinges on the fact that although iron is abundant in the sediments, it is not reactive enough for use by iron reducing microorganisms. The majority of organic carbon turnover is therefore driven by methanogenesis. The authors use this data to suggest that methanogenesis would be a significant process in ferruginous sediments, and that the extent of the contribution of methanogens was linked to Fe reactivity.

This study provides novel insights into organic matter turnover in unusually ferruginous sediments and provides some clues as to what processes may have dominated in ancient ocean environments. The paper will certainly be of interest to those studying marine sediments, and the community of researchers interested in how sedimentary and atmospheric processes may have been linked in Earth's history. It will certainly influence the thinking in the field, particularly in regard to the importance of Fe reduction in ferruginous sediments.

Most of the claims are convincing, although there are some areas where clarification or additional discussion are required:

RESPONSE: *We thank the reviewer for their detailed report. The great attention to detail helped us to improve the manuscript considerably. We hope that the reviewer will consider our manuscript to now be acceptable. Below is a detailed response to the reviewer's comments.*

There are no errors included in figure 2, 3 or 4, or figure S2, S3 or S4 which represent all of the main geochemical and incubation data. It is unclear whether they were omitted for clarity, whether they are smaller than the symbols or whether there was no replication of these measurements.

RESPONSE: *We originally excluded error bars mostly for clarity. We have now added (Table S5) to the Supplementary Information, which includes minimum, average and maximum estimates for every value plotted in Fig. 2. For the TOC and C/N data the error bars are smaller than the symbols. The figure caption now includes this information.*

In Fig. 3 the figure caption now includes information on the error.

For Fig. 4 we cannot provide error bars due to insufficient sample amounts for replicate measurements. Instead, we now include characteristic analytical figures of merit in the supplementary methods.

The rationale used to separate iron into different fractions are not always clearly explained. I have listed a number of queries regarding this below. There are also two potential gaps in the data set in that both Fe mineralogy and microbial processes are always inferred, but never directly observed. To strengthen the interpretations in the manuscript it would be great if the authors could consider the possibility of incorporating the following analyses:

Fe mineralogy: Since the Fe mineralogy is so central to the story it would significantly improve the manuscript to have some direct measurement. The sequential extractions are fine (and a lot of effort which I appreciate!), but they do not give any real specificity on what the mineral content actually is. Given the high iron content, these seem like prime candidates for Moessbauer spectroscopy.

RESPONSE: *While we appreciate the value of mineralogical information, we want to stress here that the precise mineralogy of the different pools of Fe extracted by each step of the Fe-speciation protocol is not the focus, rather crucially; it is the reactivity of these different pools towards microbial and chemical reduction that is important. For example, once step 3 of the Fe-speciation extraction (Fe_{dith}) is reached, the Fe extracted in this fraction operationally represents crystalline oxide phases (almost certainly Fe(III), given that the chemical mode of action for the dithionite is reductive dissolution). Regardless of what these phases actually are, they are by definition reducible. Thus, the observation that these dithionite reducible Fe(III) phases persist under such conditions, is highly important. Nevertheless, we have tethered our extraction data to underlying Fe mineralogy through XRD analyses, which confirm the presence of abundant goethite throughout the sediment studied. We attempted to make these points clearer in the Main Text and Supplementary Information.*

Line 172: *The mineralogical composition of dithionite extracted Fe(III) was further evaluated using quantitative X-ray diffraction, which identified the main Fe(III) bearing phase throughout the core as goethite (Fig. S4 and Table S6).*

Line 430: *Sample masses of 200 mg of sediment were weighed into 15 mL centrifuge tubes, and subjected to the Fe-speciation sequential extraction scheme based on the protocol of Poulton and Canfield (Poulton and Canfield, 2005):*

- *Leaching the sample with 0.5 N HCl for 1 h extracted the highly reactive Fe in hydrous ferric oxides like ferrihydrite or lepidocrocite (Fe_{HCl}). Iron speciation was determined spectrophotometrically (Thamdrup et al.,*

1994; Viollier *et al.*, 2000). However, easily reducible ferric iron within the Fe_{aca} fraction was always below the limit of detection in all samples, so that this fraction entirely consists of ferrous iron.

- $Fe(II)$ in carbonate and poorly crystalline phases was extracted by leaching the sediment with a sodium acetate solution adjusted with acetic acid to pH 4.5 with acetic acid for 48 h (Fe_{aca}).
- Reactive $Fe(III)$ in goethite and hematite was extracted by treating the sediment sample with a sodium dithionite solution (50 g L^{-1}) buffered to pH 4.8 with 0.35 M acetic acid/0.2 M sodium citrate for 2 h (Fe_{dith}). Due to the reductive dissolution in this step, the resulting extract only contains $Fe(II)$. However, as the main minerals dissolved in this step only contain $Fe(III)$ we consider this pool to consist only of $Fe(III)$. We further verified the selectivity of the extraction by performing a dithionite leaching experiment on two samples from the upper 30 cm of the Towuti sediment. We first quantified via qXRD the amount of goethite in these samples, and then subsequently leached them with dithionite. Our results (Table S6) demonstrate that within error of the qXRD and Fe -speciation measurements, dithionite quantitatively extracts goethite in the Lake Towuti sediments. Furthermore, XRD data demonstrates the presence of goethite throughout the core (Fig. S4), and thus we conclude that Fe contained in our dithionite extractions was indeed goethite.
- Leaching the sediment sample with 0.2 M ammonium oxalate/0.17 M oxalic acid solution (pH 3.2) for 6 h extracted the reactive Fe present in magnetite (Fe_{oxa}). Given the mixed valence of Fe in magnetite, 2/3 of the extracted Fe was counted as $Fe(III)$ and 1/3 as $Fe(II)$.
- Finally, samples were subjected to a near boiling 6 N HCl extraction for 24 h to extract the remaining unreactive Fe in silicates and Fe -bearing clays like nontronite (Fe_{sil}). $Fe(II)/Fe(III)$ ratios were not determined.

Figure S4: XRD spectra for the upper 12 m of Lake Towuti sediment. The main reflectance peaks for goethite are identified (labeled vertical grey lines “G”), which reveal goethite in every sample.

In addition, we performed a dithionite leaching experiment on two samples from the upper 30 cm of the Towuti sediment. We first quantified via qXRD the amount of goethite in these samples, and then subsequently leached them with dithionite. Our results (Table S6, below) demonstrate that within error of the qXRD and Fe -speciation measurements, dithionite quantitatively extracts goethite in the Lake

Towuti sediments. This gives us confidence in our conclusions that Fe contained in our dithionite extractions at depth, was indeed goethite. We have added this table (Table S5) and a brief discussion to the Supplementary Information.

Table S6: Iron concentrations measured by quantitative XRD (qXRD) and dithionite extractions. The data show that within error of the qXRD and Fe-speciation measurements, dithionite quantitatively extracts goethite in the Lake Towuti sediments.

Mineral	Ideal Formula	Depth Interval (1-2 cm)			Depth Interval (35-40 cm)		
		Mass (wt%)	XRD Fe (wt%)	Fe-speciation (wt%)	Mass (wt%)	XRD Fe (wt%)	Fe-speciation (wt%)
Goethite	$\alpha\text{-FeO(OH)}$	9.0	5.7	4.8	7.0	4.4	3.7

Microbial community: Some discussion of what is known about the microbial community in the core would be extremely beneficial since all interpretations rely on microbial processes.

The authors write that the core was collected specifically for Geomicrobiological analyses which suggests samples suitable for this kind of analysis would have been preserved.

I appreciate these are not minor tasks. If it is beyond the capability of the authors to explore this, perhaps there is data in the literature from other studies in Lake Towuti which would allow for some comment on this.

RESPONSE: *Here we respectfully disagree. The connections between microbial taxonomy and biogeochemical process are often tenuous, at best. Fe reduction is a case in point, given that it is spread across multiple phyla and functional markers, like those for sulfate reduction (e.g. the dsr genes) and methanogenesis, are absent. We do have abundant microbial community information (amplicon-based community profilin as well as metagenomics), but for many reasons, such as those noted above, it does not inform on the subject of the current manuscript. As a teaser, we can say that canonical Fe-reducers like Shewanella putrefaciens and Geobacter metallireducens are absent from the sediments, though close relatives to these taxa are present. Nevertheless, the facultative metabolisms characteristic of the lineages imply both may be involved in fermentation and/or syntrophy rather than the terminal oxidation pathways most central to our narrative.*

Overall, this is an interesting study where the claims are generally supported by the results. I would be very happy to see this ultimately published.

Specific comments

Line 134: Highlight in the text that siderite formation in Table 3 is modelled and not measured

RESPONSE: *We have taken the reviewers suggestion and have been more explicit here.*

Line 159: *Concomitantly high DIC and Fe(II) concentrations translate to pore waters that are supersaturated with respect to siderite (FeCO₃) throughout the sediment (Table S3) (Vuillemin et al., 2019). Despite the high abundance of acetate extractable Fe(II) and pore water supersaturation, siderite only accumulates in sporadic carbonate-rich layers (Vuillemin et al., 2019), and mineral carbon is low throughout most of the sediment (Ordoñez et al., 2019). Much of this acetate extractable Fe(II) and by extension mineral carbon and siderite likely forms in response to Fe(II) reduction in the water column (Bauer et al., 2020a), as previously shown.*

Line 139: Highlight that this is total extractable iron concentration

RESPONSE: *We have added the word “extractable”.*

Line 140: Lake Towuti's sediment has extremely high total extractable Fe concentrations (Fig. 2), with maximum values $>2500 \mu\text{mol Fe cm}^{-3}$ (20% dry wt.).

Figure 2: There is no error associated with any of the data in figure 2. The iron extractions especially can be quite variable as small differences in extraction time for each step can influence the amount of iron extracted. There is no indication in the methods how much replication was included for these extractions.

RESPONSE: We expanded the methods section in the main text, which now includes the number of replicates and several other information requested by the reviewer. There is now a table (Table S5) in the supplement with all data used in Fig. 2, including error margins.

Line 149: Clarify whether the “total iron pool” referred to here is the sum of the Fe concentration in all fractions or whether the 30% figure stated the percentage of the 0.5M HCl extractable iron.

RESPONSE: We have clarified that this refers to the total extractable iron pool.

Line 150: The Fe_{HCl} fraction comprises up to 30 % of the total extractable Fe pool. Within the Fe_{HCl} fraction, Fe(II) is abundant, while Fe(III) is below our limit of detection ($10 \mu\text{mol cm}^{-3}$) at all depths.

Line 161: Dithionite is a poor extractant for nontronite (< 5% reported by Poulton & Canfield in their original iron extraction protocol). The text should be modified to highlight this. Nontronite will show up in the fraction which was boiled in strong HCl.

RESPONSE: The reviewer is correct and we have rephrased the text accordingly.

Line 458: Finally, samples were subjected to a near boiling 6 N HCl extraction for 24 h to extract the remaining unreactive Fe in silicates and Fe-bearing clays like nontronite (Fe_{Sil}). Fe(II)/Fe(III) ratios were not determined.

Line 162: Suggest also stating “ $880 \mu\text{mol cm}^{-3}$ ” as a percentage of total iron as done for the other fractions.

RESPONSE: A percentage value is now provided.

Line 170: We therefore targeted these phases using reductive sodium dithionite extractions (Fe_{dith}) (Poulton and Canfield, 2005) and found them to be abundant in Lake Towuti's sediments, comprising up to $880 \mu\text{mol cm}^{-3}$ or 42% of the total Fe pool (Fig. 2).

Line 162: How was the value stated for dithionite-extractable iron quantified? From the text it reads like the Fe(III) in the extraction was quantified, but that would not make sense for an extraction based on a reductive dissolution i.e. all Fe should be Fe(II) in the extract regardless of its original redox state.

RESPONSE: Indeed, chemically speaking, the dithionite extraction is a reductive procedure, dissolving Fe(III) bound in oxide minerals and liberating Fe(II) in solution. When measuring the Fe-concentration in the dithionite leachates, we measure total Fe. Importantly however, all of the Fe liberated by the dithionite extraction is assumed to have initially been Fe(III) bound in crystalline oxide phases, given that as the reviewer notes, the mode of action is reductive. When we refer to Fe in the dithionite pool, we are exclusively referring to the Fe(III) concentration of the crystalline Fe(III)-oxide pool. Please see revised text in our responses above.

Line 164: The Fe_{oxa} extraction which will extract magnetite is not necessarily only contributing to the reactive Fe(III) as magnetite is mixed valent. It is not clear from the description in the methods whether both Fe(II) and Fe(III) were quantified in all extracts or only in the selected ones mentioned.

RESPONSE: *We used the stoichiometric Fe(II)/Fe(III) ratio of Magnetite ($Fe_2^{3+}Fe^{2+}O_4$) to divide the Fe pool of this fraction between the two redox states. So 2/3 is counted as Fe(III) and 1/3 as Fe(II). We added an explanation in the methods section, please see revised text in our response above.*

Fe(II) and Fe(III) concentrations were explicitly determined in the 0.5 N HCl extracts, spectrophotometrically, and only in the case of the 0.5 N HCl extractions, was both Fe(II) and Fe(III) concentrations determined explicitly. We added an explanation in the methods section. For the remaining highly reactive Fe extractions, the redox state of Fe in the target minerals are assumed as follows; Fe(II) for acetate extractable phases, and Fe(III) for crystalline dithionite extractable phases. We have made these points clearer in the revised text.

Line 166: How significant is this decrease in dithionite-extractable Fe in the upper one cm? Without any error this is hard to tell.

RESPONSE: *The error associated with our Fe-concentration measurements is now tabulated in the Supplementary Information (Table S5), showing that the decrease in dithionite extractable Fe is significant and larger than the error associated with this measurement.*

Line 172: It reads here like the authors would expect a decrease in both aqueous Fe(II) and pH when Fe reduction was occurring. Can they expand on why this is the case. The pH profiles should also be referenced here (they are not in figure 4). Can the authors expand on the extent to which it would really be expected that pH is significantly altered by Fe reduction in such a well-buffered system?

RESPONSE: *We rephrased the sentence. The pH profiles are in figure 4 on the far right panel, so we are unclear as to what the reviewer is looking for here.*

Line 183: *The apparent lack of Fe(III) reduction in much of the sediment is consistent with the pore water profiles of Fe²⁺ concentration (Fig. 3) and pH (Fig. 4), the latter would be expected to increase in response to appreciable Fe(III) reduction as this is a proton-consuming process (Froelich et al., 1979).*

Figure 4: How do the authors explain the coinciding increase of Fe²⁺ and acetate, and decrease in lactate at around 4-6m. Considering that iron reducers convert lactate to acetate and Fe(III) to Fe(II), is this not at least circumstantial evidence for iron reduction here?

RESPONSE: *We do not completely exclude iron reduction in Lake Towuti sediment, but the rates observed are low enough that they do not play any quantitative role. The coinciding increase of Fe²⁺ and decrease in lactate is a wonderful example of this, the maximum Fe²⁺ production (i.e. Fe(III) reduction) rate is about 1 pmol cm⁻³ d⁻¹ and the concomitant lactate consumption rate is 0.05 pmol cm⁻³ d⁻¹. These rates are dwarfed by the the methanogenesis rates of >1,000 pmol cm⁻³ d⁻¹.*

Line 192-193: In general I think the evidence is quite strong that Fe reduction rates are low, but I am not entirely convinced (yet) that the Fe reduction is entirely limited to the surface sediment due to the observations in my comment on Figure 4. It also seems to me that the claim that the Fe(III) minerals are stable over tens of thousands of years is a bit of a stretch based on the available data which simply compares how reactive the minerals there are, but cannot show whether these are the same type of mineral.

RESPONSE: See previous response regarding Figure 4. Furthermore, we now show that the mineral goethite is stable throughout the core.

Line 202: The authors highlight here that whilst the geochemical data and modelling do not point to sulfate reduction, the radiotracer experiments show there is sulfate reduction. When their claim that iron reduction does not occur at depth is exclusively based on geochemical data, can they rule out a similar process for iron?

RESPONSE: For clarification, we do not say that iron reduction does not occur, but rather that it only plays a minor role. In the main text we claim that the sulfate reduction rates measured by radiotracer are potential rates and should therefore be taken “as an indication for metabolic potential only”. By analogy, our geochemical data show that Fe reduction rates are very low in comparison to methanogenesis. The metabolic potential for Fe reduction may, indeed, be high, but we have no information on this. Unlike sulfate reduction, which can be measured through radiotracer experiments, no such technique exists to determine metabolic potential for Fe reduction and we thus cannot rule it out. Nevertheless, this is not important for our conclusions, which are based on actual process rather than potential for it.

Line 263: Assuming a 2:1 stoichiometry for the conversion of organic matter to methane, methanogenesis accounts for the conversion of $440 \pm 170 \text{ mmol m}^{-2} \text{ yr}^{-1}$ organic carbon. This rate far exceeds those of all other carbon mineralization processes combined but is still less than, and therefore well supported by, the carbon accumulation rate. Within the upper 12 m of sediment, sulfate reduction, iron reduction, as well as methanogenesis combined add up to a total organic carbon mineralization rate of $519 \pm 191 \text{ mmol m}^{-2} \text{ yr}^{-1}$, with methanogenesis being the dominant process (85 - 92%) followed by iron reduction (8 %) and sulfate reduction (<1-7 %).

Line 278: Our data demonstrate that methanogenesis is the dominant (>85%) pathway for carbon mineralization in Lake Towuti sediment. pSRR rates suggest that sulfate reduction may occur throughout the upper 12 m of sediment, but here it makes a minor, even insignificant, contribution to total organic carbon mineralization. Fe-reduction, likewise, is active in the water column and uppermost sediments of Lake Towuti, where it drives formation of primary authigenic magnetite (Bauer et al., 2020b) and other Fe(II) containing minerals (Zegeye et al., 2012), but remarkably it plays a minor role in the sediments despite the high abundance of Fe(III)-containing minerals.

Line 278: The authors suggest crystallinity of the Fe minerals leads to inhibited Fe reduction and the out-competition of Fe-reducers by methanogens, but can it be ruled out that the availability of potential electron donors for iron reduction does not also play a role? Fe-reducers and methanogens have some substrates in common (e.g. acetate, H₂) but their flexibility towards other substrates is different. The section would benefit from some discussion of other potential drivers of the lack of Fe reduction, in addition to the crystallinity.

RESPONSE: In Fig. 3 we present pore water concentration profiles of formate, acetate, lactate and butyrate, all indicating active fermentation throughout the core. These four VFAs plus hydrogen (which we could not measure but consider to be present) provide a broad range of potential substrates. While we agree with the reviewer that some organisms can only utilize a narrow range of electron donors, those five substrates should be sufficient to cover most substrate requirements. Furthermore, Fe reducers, as the reviewer notes, are more flexible in their electron donor preferences than methanogens, which thus should favor Fe reduction, opposite to our observations.

Line 839-849: The authors should supply additional information here on how the Fe(II) and Fe(III) were calculated.

RESPONSE: *Please see responses above. We have added a table to the Supplementary Information with detailed results from the sequential extraction (Table S5) and we revised the methods section in the main text.*

REFERENCES

- Bauer, K., Byrne, J., Kenward, P., Simister, R., Michiels, C., Friese, A., Vuillemin, A., Henny, C., Nomosatryo, S., and Kallmeyer, J., 2020a, Magnetite biomineralization in ferruginous waters and early Earth evolution: *Earth and Planetary Science Letters*, v. 549, p. 116495.
- Bauer, K. W., Byrne, J. M., Kenward, P., Simister, R. L., Michiels, C. C., Friese, A., Vuillemin, A., Henny, C., Nomosatryo, S., Kallmeyer, J., Kappler, A., Smit, M. A., Francois, R., and Crowe, S. A., 2020b, Magnetite biomineralization in ferruginous waters and early Earth evolution: *Earth and Planetary Science Letters*, v. 549, p. 116495.
- Beulig, F., Røy, H., McGlynn, S. E., and Jørgensen, B. B., 2019, Cryptic CH₄ cycling in the sulfate–methane transition of marine sediments apparently mediated by ANME-1 archaea: *The ISME Journal*, v. 13, no. 2, p. 250-262.
- Canfield, D. E., Zhang, S., Wang, H., Wang, X., Zhao, W., Su, J., Bjerrum, C. J., Haxen, E. R., and Hammarlund, E. U., 2018, A Mesoproterozoic iron formation: *Proceedings of the National Academy of Sciences*, v. 115, no. 17, p. E3895-E3904.
- Froelich, P. N., Klinkhammer, G. P., Bender, M. L., Luedtke, N. A., Heath, G. R., Cullen, D., Dauphin, P., Hammond, D., Hartman, B., and Maynard, V., 1979, Early oxidation of organic matter in pelagic sediments of the eastern equatorial Atlantic: suboxic diagenesis: *Geochimica et Cosmochimica Acta*, v. 43, p. 1075-1090.
- Ordoñez, L., Vogel, H., Sebag, D., Ariztegui, D., Adatte, T., Russell, J. M., Kallmeyer, J., Vuillemin, A., Friese, A., and Crowe, S. A., 2019, Empowering conventional Rock-Eval pyrolysis for organic matter characterization of the siderite-rich sediments of Lake Towuti (Indonesia) using End-Member Analysis: *Organic Geochemistry*, v. 134, p. 32-44.
- Poulton, S. W., and Canfield, D. E., 2005, Development of a Sequential Extraction Procedure for Iron: Implications for Iron Partitioning in Continentally Derived Particulates: *Chemical Geology*, v. 214, p. 209–221.
- Poulton, S. W., and Canfield, D. E., 2011, Ferruginous conditions: a dominant feature of the ocean through Earth's history: *Elements*, v. 7, no. 2, p. 107-112.
- Siahi, M., Tsikos, H., Rafuza, S., Oonk, P. B., Mhlanga, X. R., van Niekerk, D., Mason, P. R., and Harris, C., 2020, Insights into the processes and controls on the absolute abundance and distribution of manganese in Precambrian iron formations: *Precambrian Research*, v. 350, p. 105878.
- Thamdrup, B., Fossing, H., and Jørgensen, B. B., 1994, Manganese, iron and sulfur cycling in a coastal marine sediment, Aarhus Bay, Denmark: *Geochimica Et Cosmochimica Acta*, v. 58, p. 5115-5129.
- Thompson, K. J., Kenward, P. A., Bauer, K. W., Warchola, T., Gauger, T., Martinez, R., Simister, R. L., Michiels, C. C., Llíros, M., Reinhard, C. T., Kappler, A., Konhauser, K. O., and Crowe, S. A., 2019a, Photoferrotrophy, deposition of banded iron formations, and methane production in Archean oceans: *Science Advances*, v. 5.
- , 2019b, Photoferrotrophy, deposition of banded iron formations, and methane production in Archean oceans: *Science Advances*, v. 5, no. 11, p. eaav2869.
- Viollier, E., Inglett, P. W., Hunter, K., Roychoudhury, A. N., and Van Cappellen, P., 2000, The ferrozine method revisited: Fe(II)/Fe(III) determination in natural waters: *Applied Geochemistry*, v. 15, no. 6, p. 785-790.

- Vuillemin, A., Wirth, R., Kemnitz, H., Schleicher, A. M., Friese, A., Bauer, K. W., Simister, R., Nomosatryo, S., Ordoñez, L., Ariztegui, D., Henny, C., Crowe, S. A., Benning, L. G., Kallmeyer, J., Russell, J. M., Bijaksana, S., Vogel, H., and Towuti Drilling Project Science Team, t., 2019, Formation of diagenetic siderite in modern ferruginous sediments: *Geology*, v. 47 no. 6, p. 540-544.
- Zegeye, A., Bonneville, S., Benning, L. G., Sturm, A., Fowle, D. A., Jones, C., Canfield, D. E., Ruby, C., MacLean, L. C., Nomosatryo, S., Crowe, S. A., and Poulton, S. W., 2012, Green rust formation controls nutrient availability in a ferruginous water column: *Geology*, v. 40, no. 7, p. 599-602.

REVIEWERS' COMMENTS

Reviewer #1 (Remarks to the Author):

The authors did their due diligence in responding to the extensive comments from reviewer #2. I believe they were able to provide a reasonable rebuttal to all the criticisms. In light of this, and given that I was already in favor of publication, I am even more so now.

Reviewer #2 (Remarks to the Author):

I have read the revised manuscript several times again, and I have no further critical comments to make in the form of an additional review. The authors have gone to great lengths in rebutting many of my earlier concerns and criticisms, and in making substantial revisions to their earlier manuscript where full rebuttal was not warranted. This has resulted in a much improved and stronger manuscript that conveys a couple of very important and pertinent new findings that deserve publication in a high-profile journal. To this end, I will only make two complementary comments below, which I feel are important for the authors to at least consider before final publication of their work. In other words, I am not suggesting any further revisions as such, but am rather making a couple of points that the authors may wish to take on-board as they make further progress with this line of their research.

1) The first point concerns the sedimentary candidates of high iron content in the oceanic environment of deep time. In my earlier review, I challenged the notion of ferruginous shales being the main mode of high iron deposition in the Neoproterozoic and Paleoproterozoic marine realm, by comparison to the classic BIF. In raising the above issue, I erroneously misquoted a paper of Poulton and Canfield (2007) instead of the one by the same authors published in 2011 and cited in the current manuscript. I therefore apologize profusely for that oversight, and kindly request a correction to it should my earlier review be published. I do, however, continue to stand my ground on the ferruginous shale versus BIF argument. The pre-GOE geological record has substantial BIF deposits globally, but I personally have not encountered any clastic sedimentary rocks that would fulfil the "ferruginous shale" definition that the authors quote. In fact, I am not entirely sure what that definition is, if any. There are sulphidic shales of all ages that may run up to 20wt% modal pyrite or more, and thus bulk-rock iron levels commensurate with a true BIF; would such rocks compositionally qualify? Alternatively, would the ferruginous shales of the authors expected to be magnetitic as BIFs predominantly are? I should stress that the S-bands in the Hamersley BIF are only a lesser component in the iron-rich stratigraphy and, to my knowledge at least, they are not even conclusively interpreted as ferruginous siliciclastic sediments. Even if they were though, who is to say that they do not represent episodically deposited muds or silts (possibly as parts of volcanic ash falls even) that have received background chemical deposition of iron exactly as in the case of the BIF sequences that contain them? In other words, the provenance of iron (and organic carbon, where relevant) is in its own right an important part of the overall discussion as I will further content in my second point below. For now, all I want to conclude for this first point is that I still do not know what these ferruginous shales exactly represent in the Precambrian geological record. Therefore, I will personally continue to consider them as a kind of a misnomer until at least one study emerges that considers such rocks from anywhere in the world directly and holistically, and establishes them as the predominant mode of iron-rich deposition in the early Precambrian by comparison to BIF.

2) My second and final point is arguably a more important one in a broader scientific sense. It concerns the two main messages conveyed in this manuscript: starting with the first, it is effectively the one promoted heavily by the authors through their chosen title and abstract, and concerns the dominance of methanogenesis in the sediments studied by the authors, and the implications thereof for the earliest oceans and climate. Although I have no major reason to contest that line of interpretation, I would like to alert the authors to the fact that the sediments of lake Towuti must be receiving a high input of iron and organic carbon from the surrounding hinterland in the form of terrigenous detritus. Hence, much of the cycling of Fe and C in Lake Towuti relates to those inputs and the reactivities of the corresponding mineral species, and not exclusively to biochemical processes

occurring intrinsically in lake waters *sensu stricto*. In other words, the primary and diagenetic processes that pertain to lake Towuti implicate a very strong terrigenous clastic component in terms of both Fe and organic matter. This, in my opinion, cannot readily find analogy to the Precambrian environment whereby the long-term source of iron in the ocean is thought to be predominantly (if not exclusively) hydrothermal, while the postulated absence of life in the terrestrial realm would preclude any measurable terrestrial organic carbon fluxes to the global ocean of that time. In other words, the overall residence times and balance of inputs and outputs of iron and organic carbon in the ocean-land system of deep time, is hardly analogous to what observed at lake Towuti. On the basis of the above, I continue to believe that the micro-environments that the Indonesian Lakes represent are not necessarily the best candidates to permit direct and meaningful comparisons with the early ferruginous oceans of the Precambrian, but in the absence of better modern analogues, they arguably still have some useful things to tell us.

The second message that the authors convey, however, is one that is somewhat downplayed and consequently remains a bit cryptic to the reader. It is specifically contained in a couple of excerpts from their manuscript, namely in lines 195-197 and 377-381. The overall message here is that water column redox cycling of Fe must have been both active and very efficient in not only reducing much of the primary trivalent iron (whatever its source) back into the water column as ferrous iron, but also in leading to the saturation and primary precipitation of reduced (or mixed-valence) iron minerals such as magnetite, carbonates, or even silicates. As far as I am concerned, this second message is by far the most important - almost as a paradigm shift - in the context of the origin of BIF as archives of early Earth evolution, and the one that I would personally have strived to emphasize if I had the privilege to lead such a study. Nevertheless, the message is still there for those who will read the final paper with the attention it deserves, so I have no real reason to "force this issue" any further, especially at this stage of this review.

Hari Tsikos

Reviewer #3 (Remarks to the Author):

I am satisfied that the authors have done a good job of addressing my questions and I think their arguments are robust. I particularly appreciate the addition of the XRD as the observation that the main oxide is goethite is helpful for putting the observations in context. It should be remembered that the susceptibility to dithionite reduction is just a proxy for microbial reducibility (likely to be less) but I agree these extractions are likely the best proxy we have available. I would recommend publication.

Response to Reviewer #2, Hari Tsikos

We would like to thank Dr. Tsikos for his thoughtful revision of our manuscript, which helped us to improve it greatly. Below we would like to respond to the two remaining complementary comments:

The first point concerns the sedimentary candidates of high iron content in the oceanic environment of deep time. In my earlier review, I challenged the notion of ferruginous shales being the main mode of high iron deposition in the Neoarchaeon and Paleoproterozoic marine realm, by comparison to the classic BIF. In raising the above issue, I erroneously misquoted a paper of Poulton and Canfield (2007) instead of the one by the same authors published in 2011 and cited in the current manuscript. I therefore apologize profusely for that oversight, and kindly request a correction to it should my earlier review be published.

RESPONSE: *We support Prof. Tsikos request to modify the original review and therefore request that this be done if consistent with the editorial policies at Nature Communications. From our perspective there is nothing that should prevent making such a correction, or rather deleting the respective section entirely. For the time being we will not change anything and leave the decision to Dr. Frischkorn.*

I do, however, continue to stand my ground on the ferruginous shale versus BIF argument. The pre-GOE geological record has substantial BIF deposits globally, but I personally have not encountered any clastic sedimentary rocks that would fulfil the "ferruginous shale" definition that the authors quote.

RESPONSE: *We apologize for the ongoing confusion associated with our definition of "ferruginous shale". Simply put, when we refer to ferruginous sediments, we are using the operational geochemical definition of ferruginous, and very straight forwardly this refers to ANY sediments deposited under anoxic conditions such that $Fe_{Pyr}/Fe_{HR} < 0.70$. In other words, any sediments where unpyritized/sulphidized highly reactive Fe persists. Our point is simply that such sediments need not fulfil the lithologic definition of BIFs, and in fact, there are many anoxic sediments with $Fe_{Pyr}/Fe_{HR} < 0.70$ deposited between 3.0 and 0.8 Ga, that were indeed not BIFs. To help clarify, we have added to following statement to the manuscript:*

Lns 122-123 "- which are by definition Fe-rich and contain highly reactive Fe that is not pyritized –"

Furthermore, some geologic descriptions taken directly from the literature of non-BIF ferruginous Precambrian sediments deposited between 2.5 and 0.5 Ga and that have abundant non-pyritized reactive Fe are described as follows; marine mudstones intercalated with shales and siltstones, shales and carbonates, shallow marine shales¹. We have revised the Main Text by removing any reference to a specific rock type.

Ln 122 – "Precipitation of Fe from these oceans resulted in the widespread deposition of ferruginous sediments and, in cases, nearly pure chemical sediments like many banded iron formations (BIFs)⁸.

In fact, I am not entirely sure what that definition is, if any. There are sulphidic shales of all ages that may run up to 20wt% modal pyrite or more, and thus bulk-rock iron levels commensurate with a true BIF; would such rocks compositionally qualify?

RESPONSE: *As above, we mean any sediment type that fulfills the geochemical definition of ferruginous ($Fe_{HR}/Fe_{Tot} > 0.38$ and $Fe_{Pyr}/Fe_{HR} < 0.70$).*

Alternatively, would the ferruginous shales of the authors expected to be magnetitic as BIFs predominantly are?

RESPONSE: *This would depend on the dominant form of highly reactive Fe that persists non-pyritized (eg., goethite vs. siderite). However, discussion of this point is beyond the scope of the present manuscript.*

I should stress that the S-bands in the Hamersley BIF are only a lesser component in the iron-rich stratigraphy and, to my knowledge at least, they are not even conclusively interpreted as ferruginous siliciclastic sediments. Even if they were though, who is to say that they do not represent episodically deposited muds or silts (possibly as parts of volcanic ash falls even) that have received background chemical deposition of iron exactly as in the case of the BIF sequences that contain them? In other words, the provenance of iron (and organic carbon, where relevant) is in its own right an important part of the overall discussion as I will further content in my second point below.

RESPONSE: From the reviewer

“Even if they were though, who is to say that they do not represent episodically deposited muds or silts (possibly as parts of volcanic ash falls even) that have received background chemical deposition of iron exactly as in the case of the BIF sequences that contain them?”

This is correct and partly the point. These are not purely chemical sediments (our end-member BIF) and if these layers have $Fe_{HR}/Fe_{Tot} > 0.38$ and $Fe_{Pyr}/Fe_{HR} < 0.70$ they would be considered ferruginous and are thus some of the ‘other’ sediments we refer to.

For now, all I want to conclude for this first point is that I still do not know what these ferruginous shales exactly represent in the Precambrian geological record. Therefore, I will personally continue to consider them as a kind of a misnomer until at least one study emerges that considers such rocks from anywhere in the world directly and holistically, and establishes them as the predominant mode of iron-rich deposition in the early Precambrian by comparison to BIF.

RESPONSE: *Prof. Tsikos raises a good point here and this likely represents an important opportunity for future research. As is often the case, rocks considered to be ‘more mundane’ often receive less attention than those that are spectacular like BIFs. There is clearly a rich archive of non-BIF Archean shales (see for example the compilation in Box 1 of the Lyons et al. 2014 review in Nature), and these would represent a prime target for Fe-speciation analyses. Field-based studies to further map and quantify shales in the large Archean terrains would also seem important.*

2) My second and final point is arguably a more important one in a broader scientific sense. It concerns the two main messages conveyed in this manuscript: starting with the first, it is effectively the one promoted heavily by the authors through their chosen title and abstract, and concerns the dominance of methanogenesis in the sediments studied by the authors, and the implications thereof for the earliest oceans and climate. Although I have no major reason to contest that line of interpretation, I would like to alert the authors to the fact that the sediments of lake Towuti must be receiving a high input of iron and organic carbon from the surrounding hinterland in the form of terrigenous detritus. Hence, much of the cycling of Fe and C in Lake Towuti relates to those inputs and the reactivities of the corresponding mineral species, and not exclusively to biochemical processes occurring intrinsically in lake waters sensu stricto. In other words, the primary and diagenetic processes that pertain to lake Towuti implicate a very strong terrigenous clastic component in terms of both Fe and organic matter. This, in my opinion, cannot readily find analogy to the Precambrian environment whereby the long-term source of iron in the ocean is thought to be predominantly (if not exclusively) hydrothermal, while the postulated absence of life in the terrestrial realm would preclude any measurable terrestrial organic carbon fluxes to the global ocean of that time. In other words, the overall residence times and balance of inputs and outputs of iron and organic carbon in the ocean-land system of deep time, is hardly analogous to what observed at lake Towuti. On the basis of the above, I continue to believe that the micro-environments that the Indonesian Lakes represent are not necessarily the best candidates to permit direct and meaningful comparisons with the early ferruginous oceans of the Precambrian, but in the absence of better modern analogues, they arguably still have some useful things to tell us.

RESPONSE: *Here, the reviewer raises valid and interesting points. However, as we stated in our previous response, we do not claim that Lake Towuti itself is a perfect or direct analogue of the pre-GOE ocean, but rather consider the fundamental biogeochemical processes operating within the lake*

to be analogous and thus extensible. We don't know how much of the sedimentary organic matter in Lake Towuti is derived from land and how much is produced in the lake but this does not matter, as it is the reactivity that counts. Figure 2 shows that the C/N ratio of the sedimentary organic matter, which is a widely accepted parameter for bulk organic matter reactivity, is between 10 and 30. For a lake, this is rather low and approximates values found in continental shelf sediments, the same oceanographical setting where BIFs formed. While we don't know much about the original reactivity of organic matter at the time of BIF deposition, we know that the material deposited in modern Lake Towuti has a similar reactivity as modern marine organic matter in intermediate water depths.

We make, and have made, similar arguments for the Fe minerals. The study by Sheppard et al. (2019) shows that the iron minerals that enter the lake undergo (or have undergone) extensive authigenic/aquatic reworking since their formation in surrounding soils. As we state in lines 370-373: "Prior studies at Lake Towuti, however, suggest that regardless of source, sedimentary iron (oxyhydr)oxides are poorly crystalline and may have experienced strong reworking prior to burial²—such conversion to authigenic phases would likely render them more reactive towards reduction." We further speculate that surface passivation through Fe(II) sorption might be the cause of this lack of reactivity. We feel that we clearly communicate the commonalities and differences between modern Lake Towuti and the pre-GOE ocean to allow a critical extension of our results to the past and therefore we elect not to change the manuscript further in this regard.

The second message that the authors convey, however, is one that is somewhat downplayed and consequently remains a bit cryptic to the reader. It is specifically contained in a couple of excerpts from their manuscript, namely in lines 195-197 and 377-381. The overall message here is that water column redox cycling of Fe must have been both active and very efficient in not only reducing much of the primary trivalent iron (whatever its source) back into the water column as ferrous iron, but also in leading to the saturation and primary precipitation of reduced (or mixed-valence) iron minerals such as magnetite, carbonates, or even silicates. As far as I am concerned, this second message is by far the most important - almost as a paradigm shift - in the context of the origin of BIF as archives of early Earth evolution, and the one that I would personally have strived to emphasize if I had the privilege to lead such a study. Nevertheless, the message is still there for those who will read the final paper with the attention it deserves, so I have no real reason to "force this issue" any further, especially at this stage of this review.

RESPONSE: *We thank the reviewer for recognizing that some of our observations may be "paradigm shifting". We agree with the reviewer that water column redox cycling of Fe is very efficient in the Malili Lakes, and is a process that warrants additional study. We have made minor revisions to our manuscript to emphasize the importance of water column processes as follows:*

Lns 361-365 "This observation emphasizes the role of primary water column processes in controlling sediment mineralogy in ferruginous environments and implies that the mineral products of Fe-respiration have strong potential to record primary water column signals."

Lns 464-465 "... and strongly implicates primary processes in setting the mineralogy of BIFs and other ferruginous sediments."